# Robust On-Policy Sampling for Data-Efficient Policy Evaluation in Reinforcement Learning

**Rujie Zhong**[1], **Duohan Zhang**[2,*], **Lukas Schäfer**[1], **Stefano V. Albrecht**[1], **Josiah P. Hanna**[3,*]

[1] School of Informatics, University of Edinburgh
[2] Statistics Department, University of Wisconsin – Madison
[3] Computer Sciences Department, University of Wisconsin – Madison
* Correspondence to {dzhang357, jphanna}@wisc.edu

## Abstract

Reinforcement learning (RL) algorithms are often categorized as either on-policy or off-policy depending on whether they use data from a target policy of interest or from a different behavior policy. In this paper, we study a subtle distinction between on-policy *data* and on-policy *sampling* in the context of the RL sub-problem of policy evaluation. We observe that on-policy sampling may fail to match the expected distribution of on-policy data after observing only a finite number of trajectories and this failure hinders data-efficient policy evaluation. Towards improved data-efficiency, we show how non-i.i.d., off-policy sampling can produce data that more closely matches the expected on-policy data distribution and consequently increases the accuracy of the Monte Carlo estimator for policy evaluation. We introduce a method called *Robust On-Policy Sampling* and demonstrate theoretically and empirically that it produces data that converges faster to the expected on-policy distribution compared to on-policy sampling. Empirically, we show that this faster convergence leads to lower mean squared error policy value estimates.

## 1 Introduction

Reinforcement learning (RL) algorithms are often categorized using the dichotomy of on-policy versus off-policy. On-policy algorithms learn about a particular target policy using data collected by behaving according to the target policy. Off-policy algorithms use data collected by behaving according to a different *behavior* policy. We study a subtle distinction between on-policy *data* versus on-policy *sampling* as a step towards more data-efficient RL algorithms. To better understand this distinction, consider a simple example. In this example, a certain target policy repeatedly visits a state in which it takes action A with probability 0.2 and action B with probability 0.8. Under on-policy sampling, after five visits to this state, we might actually observe action A 2 times and action B 3 times instead of the expected 1 and 4 times. Alternatively, we could collect data off-policy by deterministically tracking the expected target policy action proportions; doing so results in observing the exact expected action frequencies. Though the latter case uses off-policy sampling, it produces data that is arguably more on-policy than the data produced by on-policy sampling.

In this paper, we study the distinction between on-policy sampling and on-policy data in the context of the RL sub-problem of policy evaluation [Zinkevich et al., 2006]. In policy evaluation, we are given an *evaluation policy* and asked to estimate the expected return that would be accrued when running the evaluation policy on a task of interest. This problem is important for high confidence deployment of RL-trained policies. In RL applications, such as robotics, data-efficient policy evaluation is of the utmost importance – we desire the most accurate estimate with minimal collected data. While much research has gone into how to most efficiently use a set of already collected data, i.e., the off-policy policy evaluation problem [Jiang and Li, 2016, Thomas and Brunskill, 2016], an implicit assumption

in the RL community is that on-policy data is preferred to off-policy data when available. When data can be collected on-policy, we can use the Monte Carlo estimator which computes a mean return estimate using trajectories sampled i.i.d. by running the evaluation policy. In the limit, with infinite trajectories, the empirical proportion of each trajectory will converge to its true probability under the evaluation policy and the estimate will converge to the true expected return. However, for any *finite* sample-size, the empirical proportion of each trajectory will likely fail to match the true probability and the estimate will have error. Such *sampling error* is an inevitable feature of i.i.d. sampling. The probability of each new trajectory is unaffected by the trajectories occurring in the past and thus the only way to ensure the empirical distribution matches the true probability is to sample a large enough data set. That is, it is only in the limit that on-policy sampling produces exactly on-policy data.

The observations made so far raise the question: "can non-i.i.d., off-policy trajectory sampling cause the empirical distribution of trajectories to converge to the expected on-policy distribution faster?" We answer this question affirmatively by introducing a method that adapts the data collecting behavior policy to consider what data has already been collected when selecting future actions. We call this method *Robust On-Policy Sampling* (ROS)[1] since the empirical distribution of data it produces converges faster to the expected on-policy trajectory distribution compared to standard on-policy sampling. We give a theoretical result supporting this claim and then confirm our theory with policy evaluation experiments in finite and continuous-valued state- and action-space domains showing 1) ROS reduces sampling error in finite datasets and 2) consequently lowers the MSE of policy value estimates compared to i.i.d. on-policy sampling.

Our paper contributes to the field of RL on two fronts. On one front, we introduce a practical method for data collection and demonstrate empirically that it leads to more accurate policy evaluation compared to on-policy sampling. Simultaneously, our work examines nuance in the on-policy versus off-policy dichotomy. A better understanding of this nuance opens up the possibility of designing new data collection procedures to improve the data efficiency of any RL algorithm that relies upon on-policy data.

## 2  Related Work

Data collection is a fundamental part of the RL problem. The most widely studied data collection problem is the question of how an agent should explore its environment to learn an optimal policy [Schäfer et al., 2022, Ostrovski et al., 2017, Tang et al., 2017]. In contrast to these approaches, our work focuses on the question of how an agent should collect data to *evaluate* a fixed policy. When given a choice of how to collect data for policy evaluation, on-policy data collection is generally preferable to off-policy data collection [Sutton and Barto, 1998]. Notable exceptions are adaptive importance sampling (AIS) methods [Oosterhuis and de Rijke, 2020, Hanna et al., 2017, Ciosek and Whiteson, 2017, Bouchard et al., 2016, Frank et al., 2008] and quasi-Monte Carlo methods [Arnold et al., 2022]. Both these AIS methods and the Quasi-Monte Carlo method of Arnold et al. [2022] lower variance in estimates computed with future samples while our method lowers the total error in the estimate computed from both past and future samples.

In one of our experiments, we consider a setting where we already have some data (collected off-policy) and must decide how to collect additional data for policy evaluation. This problem has been previously studied in the bandit literature [Tucker and Joachims, 2022] or when there are only a finite number of policies that could be ran [Konyushova et al., 2021]. These prior works also show that on-policy data collection is a sub-optimal choice. They differ from (and are complementary to) our work in that they still use i.i.d. sampling for data collection whereas we show how non-independent sampling can be used to produce data that more closely matches a desired distribution.

The method we introduce in this paper is motivated by the idea of decreasing sampling error in all collected data. Previous work has considered how sampling error can be reduced *after* data collection by re-weighting the obtained samples. For example, Hanna et al. [2021] show how importance sampling with an estimated behavior policy can lower sampling error and lead to more accurate policy evaluation. Similar methods have also been studied for policy evaluation in multi-armed bandits [Narita et al., 2019, Li et al., 2015] and temporal-difference learning [Pavse et al., 2020]. These prior works assume data is available a priori and ignore the question of how to collect it when unavailable.

---

[1] We provide an open-source implementation of ROS and all experimental data at `https://github.com/uoe-agents/robust_onpolicy_data_collection`.

Finally, the idea of adapting the *sampling distribution*, (i.e., behavior policy) has analogs outside of policy evaluation in Markov decision processes. O'Hagan [1987] identifies flaws with i.i.d. sampling for Monte Carlo estimation that motivate taking past samples into account. Rasmussen and Ghahramani [2003] use Gaussian processes to represent uncertainty in an expectation to be evaluated and use this uncertainty to guide future sample generation. Concurrent to this work, Mukherjee et al. [2022] introduced an uncertainty aware method for data collection in policy evaluation that can be seen as an adaptation of these ideas to the RL.

## 3 Preliminaries

In this section, we introduce notation, formalize the policy evaluation problem, and introduce the Monte Carlo estimator for policy evaluation.

### 3.1 Notation

We assume the environment is a finite-horizon, episodic *Markov decision process* (MDP) with state set $\mathcal{S}$, action set $\mathcal{A}$, transition function, $P : \mathcal{S} \times \mathcal{A} \times \mathcal{S} \rightarrow [0, 1]$, reward function $R : \mathcal{S} \times \mathcal{A} \rightarrow \mathbb{R}$, discount factor $\gamma$, maximum horizon $l$, and initial state distribution $d_0$ [Puterman, 2014]. We assume that $\mathcal{S}$ and $\mathcal{A}$ are finite though our empirical analysis considers both settings. We assume that the transition and reward functions are unknown. A policy, $\pi : \mathcal{S} \times \mathcal{A} \rightarrow [0, 1]$, is a function mapping states and actions to probabilities. We use $\pi(a|s) := \pi(s, a)$ to denote the conditional probability of action $a$ given state $s$ and $P(s'|s, a) := P(s, a, s')$ to denote the conditional probability of state $s'$ given state $s$ and action $a$. Since we assume a finite-horizon, we assume the state definition implicitly includes temporal information [Agarwal et al., 2022].

Let $h := (s_0, a_0, r_0, s_1, \ldots, s_{l-1}, a_{l-1}, r_{l-1})$ be a *trajectory* and $g(h) := \sum_{t=0}^{l-1} \gamma^t r_t$ be the *discounted return* of $h$. Any policy induces a distribution over trajectories, $\Pr(h|\pi)$. We define the *value* of a policy, $v(\pi)$, as the expected discounted return when sampling a trajectory by following policy $\pi$: $v(\pi) := \mathbf{E}[g(H)|H \sim \pi] = \sum_h \Pr(h|\pi)g(h)$ where $H$ is a random variable representing a trajectory and $H \sim \pi$ denotes sampling $H$ by running $\pi$ in the given environment.

### 3.2 Policy Evaluation

In the policy evaluation problem, we are given an *evaluation policy*, $\pi_e$, for which we would like to estimate $v(\pi_e)$. Conceptually, algorithms for policy evaluation involve two steps: collecting data (or receiving previously collected data) and computing an estimate from that data. We assume that data is collected by running a policy which we call the *behavior policy*. If the behavior policy is the same as the evaluation policy data collection is *on-policy*; otherwise it is *off-policy*. Whether on-policy or off-policy, we assume the data collection process produces a set of trajectories, $D := \{H_i\}_{i=1}^n$ and write $D \sim \pi$ to denote collecting these trajectories by running policy $\pi$. The final value estimate is then computed by a policy evaluation estimator (PE) that maps the set of trajectories to a scalar-valued estimate of $v(\pi_e)$. Following earlier work in policy evaluation (e.g., [Thomas and Brunskill, 2016, Jiang and Li, 2016]), we set our goal to be policy evaluation with low *mean squared error* (MSE):

$$\mathrm{MSE}\Big[\mathrm{PE}\Big] := \mathbf{E}\Big[\Big(\mathrm{PE}(D) - v(\pi_e)\Big)^2 \,\Big|\, D \sim \pi_b\Big], \tag{1}$$

where $\pi_b$ is the behavior policy that is run to collect $D$ and PE is a generic policy evaluation estimator.

### 3.3 Monte Carlo Policy Evaluation

Perhaps the most fundamental, model-free policy evaluation method is the *Monte-Carlo* (MC) estimator. Given a data set, $D$, of $n$ trajectories, the Monte Carlo estimate, $\mathrm{MC}(D)$, is the mean return over $D$:

$$\mathrm{MC}(D) := \frac{1}{n}\sum_{i=1}^n g(H_i) = \sum_h \Pr(h|D)g(h), \tag{2}$$

where $\Pr(h|D)$ denotes the empirical probability of $h$, i.e. how often $h$ appears in $D$.

If trajectories in $D$ are collected i.i.d. by running $\pi_e$ (i.e., *on-policy* sampling), the Monte Carlo estimator is unbiased and consistent assuming $g(h)$ is bounded [Sen and Singer, 1993]. However, this method can have high variance as on-policy sampling may require many trajectories for the empirical trajectory distribution $\Pr(h|D)$ to accurately approximate $\Pr(h|\pi_e)$. Since on-policy sampling collects each trajectory i.i.d., it relies on the law of large numbers for an accurate weighting on each possible return. We call error between $\Pr(h|D)$ and $\Pr(h|\pi_e)$ *sampling error*.

## 4  Data-Conditioned Monte Carlo Estimates

In this section, we motivate how an estimator that uses on-policy data can benefit from off-policy sampling. Specifically, we consider the Monte Carlo estimator and suppose that we have already collected a data set, $\mathcal{D}_1$, of trajectories. We now wish to collect an additional set of trajectories, $D_2$, and compute the Monte Carlo estimate with the set $\mathcal{D}_1 \cup D_2$. Note that $\mathcal{D}_1$ is a fixed set (the trajectories already observed) while $D_2$ is a random variable (the trajectories yet to be observed). How should $D_2$ be collected for minimal MSE policy evaluation with the Monte Carlo estimator? Our analysis in this section suggests that i.i.d. sampling of trajectories with $\pi_e$ may be a sub-optimal choice.

In this setting, the Monte Carlo estimator using $\mathcal{D}_1 \cup D_2$ can be written as:

$$\text{MC}(\mathcal{D}_1 \cup D_2) \coloneqq \underbrace{\frac{1}{n}\sum_{i=1}^{n_{\mathcal{D}_1}} g(h_i)}_{\text{fixed value}} + \underbrace{\frac{1}{n}\sum_{i=1}^{n_{D_2}} g(H_i)}_{\text{random variable}}, \tag{3}$$

where $n_{\mathcal{D}_1}$ and $n_{D_2}$ are the number of trajectories in $\mathcal{D}_1$ and $D_2$, respectively and $n = n_{\mathcal{D}_1} + n_{D_2}$. We refer to (3) as the *data-conditioned Monte Carlo estimator*.

Viewing the Monte Carlo estimator as a sum between a fixed quantity and a random quantity changes how we view the statistical properties of the estimator. For instance, while the Monte Carlo estimator is known to be unbiased under on-policy sampling, its data-conditioned estimate is biased as shown in the following proposition.

**Proposition 1.** *The data conditioned Monte Carlo estimator is biased under on-policy sampling of $D_2$ unless $\text{MC}(\mathcal{D}_1) = v(\pi_e)$ or $\mathcal{D}_1 = \emptyset$. That is:*

$$\mathbf{E}\left[\text{MC}(\mathcal{D}_1 \cup D_2) \,\middle|\, D_2 \sim \pi_e\right] \neq v(\pi_e).$$

*Proof.* See Appendix A. $\qquad\qquad\square$

**Remark 1.** *Proposition 1 holds even if $\mathcal{D}_1$ was collected under on-policy sampling as well. When $\mathcal{D}_1$ was collected under on-policy sampling then the Monte Carlo estimator is unbiased considering all possible realizations of $\mathcal{D}_1$. However, once the trajectories in $\mathcal{D}_1$ are fixed, it no longer matters what others values they could have taken.*

Can we reduce the bias of the data-conditioned Monte Carlo estimator by collecting $D_2$ with a policy that is different than $\pi_e$? We conclude this section with an example showing that we can. Consider a one-step MDP with one state, $s$, and two actions, $a_0$ and $a_1$. The return following $a_0$ is 2 and the return following $a_1$ is 4. The evaluation policy is $\pi_e(a_0|s) = \pi_e(a_1|s) = 0.5$. Suppose that, after sampling 3 trajectories, $\mathcal{D}_1$ contains two of $\{s, a_0, 2\}$ and one occurrence of $\{s, a_1, 4\}$. Note that action $a_0$ is over-sampled relative to its true probability in $s$ and $a_1$ is under-sampled. If we collect an additional trajectory with $\pi_e$ the expected value of the Monte Carlo estimate is: $\frac{1}{4}(2 + 2 + 4 + 2\pi_e(a_0) + 4\pi_e(a_1)) = \frac{11}{4} = 2.75$. The true value, $v(\pi_e) = 3$ and thus, *conditioned on prior data*, the Monte Carlo estimate is biased in expectation as shown in Proposition 1. If instead we choose the behavior policy such that $\pi_b(a_1) = 1$ then neither action is over- or under-sampled and the expected value of the Monte Carlo estimate is the exact true value: $\frac{1}{4}(2 + 2 + 4 + 4) = \frac{12}{4} = 3$.

This example highlights that adapting the behavior policy to consider previously collected data can lower the expected finite-sample error of policy evaluation. In the next section, we introduce an adaptive data collection method that adjusts the behavior policy based on what data has already been observed so as to lower the MSE of a Monte Carlo estimate using all observed data.

# 5 Robust On-Policy Data Collection

In this section, we introduce a method that adapts the data-collecting behavior policy online to minimize sampling error in the data used by the Monte Carlo estimator. Specifically, let $\mathcal{D}_t$ denote all trajectories observed up to time-step $t$ of the current trajectory (including the partial current trajectory). At time-step $t$, our method sets the behavior policy so as to reduce the current sampling error, i.e., divergence between $\Pr(h|\pi_e)$ and $\Pr(h|\mathcal{D}_t)$. Our method can be run starting with $\mathcal{D}_t = \emptyset$ or already containing trajectories in a setting like that described in the preceding section.

To reduce sampling error when collecting future trajectories, we want to adjust the behavior policy to increase the probability of under-sampled trajectories, i.e., $h$ for which $\Pr(h|\mathcal{D}_t) < \Pr(h|\pi_e)$. Unfortunately, the trajectory distributions are unknown because the transition function, $P$, is also unknown. Instead, we will increase the probability of under-sampled actions. Let $\pi_D : \mathcal{S} \times \mathcal{A} \to [0, 1]$ denote the *empirical policy* which gives the proportion of times that each action was taken in each state in $\mathcal{D}_t$. If $\pi_D(a|s) < \pi_e(a|s)$, then $a$ has appeared less often in the data than it would in expectation under $\pi_e$. Thus, we should increase the probability of $a$ in $s$ for future data collection.

When the state and action spaces are finite, $\pi_D$ can be computed exactly as the maximum likelihood policy under $\mathcal{D}_t$:

$$\pi_D := \arg\max_\pi \mathcal{L}(\pi), \qquad \mathcal{L}(\pi) := \sum_{h \in \mathcal{D}_t} \sum_{t'=0}^{l-1} \log \pi(a_{t'}|s_{t'}), \qquad (4)$$

where the argmax is taken with respect to all policies. In larger MDPs, we require function approximation which may make $\pi_D$ hard to compute and update online as new data is collected. Fortunately, with an additional assumption we can determine the direction to adjust action probabilities without explicitly computing $\pi_D$. This assumption is that $\pi_e$ belongs to a class of differentiable, parameterized policies and is parameterized by vector $\boldsymbol{\theta} \in \mathbb{R}^d$. This assumption is mild for many RL applications as it permits tabular, linear, and neural network policy representations. We use $\boldsymbol{\theta}_e$ to represent the parameter values for $\pi_e$. We show in the next subsection that the gradient of the log-likelihood at $\boldsymbol{\theta}_e$, $\nabla_{\boldsymbol{\theta}} \mathcal{L}(\pi_{\boldsymbol{\theta}})|_{\boldsymbol{\theta}=\boldsymbol{\theta}_e}$, can be used to make sampling-error-reducing changes to the behavior policy.

## 5.1 Robust On-Policy Sampling

Our primary algorithmic contribution – **R**obust **O**n-Policy **S**ampling (ROS) – reduces sampling error by adapting the behavior policy with a single step of gradient descent on the log-likelihood at each time-step. From here on, we use $\nabla_{\boldsymbol{\theta}} \mathcal{L}$ to denote the gradient of the log-likelihood evaluated at $\boldsymbol{\theta}_e$. Observe that $\nabla_{\boldsymbol{\theta}} \mathcal{L}$ provides a direction to adjust $\boldsymbol{\theta}_e$ to *increase* the probability of actions that were over-sampled relative to their probability under $\pi_e$. Thus, $-\nabla_{\boldsymbol{\theta}} \mathcal{L}$ provides a direction to adjust $\boldsymbol{\theta}_e$ to *decrease* the probability of over-sampled actions for which $\pi_D(a|s) > \pi_e(a|s)$. With this insight, ROS is able to adapt $\boldsymbol{\theta}_e$ so that $\pi_D$ tracks $\pi_e$ without ever computing $\pi_D$. At each time-step, ROS computes $\nabla_{\boldsymbol{\theta}} \mathcal{L}$ with all state-action pairs previously observed and then changes the evaluation policy parameters with a single step of gradient descent so that under-sampled actions have greater probability than they would have under $\pi_e$.

Pseudocode for ROS is given in Algorithm 1. ROS first computes $\nabla_{\boldsymbol{\theta}} \mathcal{L}$ with previously collected trajectories if any are provided (Line 4). ROS then collects $n$ additional trajectories by interacting with the given MDP (Lines 6-14). For each action selection, ROS sets the behavior policy parameters as $\boldsymbol{\theta}_e - \alpha \nabla_{\boldsymbol{\theta}} \mathcal{L}(\pi_{\boldsymbol{\theta}})|_{\boldsymbol{\theta}=\boldsymbol{\theta}_e}$ (Lines 9 and 10). It then computes $\nabla_{\boldsymbol{\theta}} \log \pi_{\boldsymbol{\theta}}(A|s)|_{\boldsymbol{\theta}=\boldsymbol{\theta}_e}$ and updates $\nabla_{\boldsymbol{\theta}} \mathcal{L}(\pi_{\boldsymbol{\theta}})|_{\boldsymbol{\theta}=\boldsymbol{\theta}_e}$ (Lines 11 and 12). Finally, the chosen action is executed in the environment, a reward received, and the agent moves to the next state (Line 13). Importantly, note that updating $\nabla_{\boldsymbol{\theta}} \mathcal{L}$ requires per-timestep computation that is linear in the number of policy parameters and remains constant as the size of $\mathcal{D}$ grows.

## 5.2 ROS Convergence

This section develops our theoretical understanding of ROS. Due to space constraints, we defer all proofs to Appendix B. First, we show that ROS converges to the expected state visitation frequencies under $\pi_e$. Second, we show that, for a fixed state, $\pi_D(\cdot|s)$ converges to $\pi_e(\cdot|s)$ faster under ROS compared to on-policy sampling. Finally, we introduce an upper bound on the squared error between

**Algorithm 1** Robust On-Policy Sampling.

1: **Input:** Evaluation policy $\pi_e$ with parameters $\boldsymbol{\theta}_e$, step size $\alpha$, previously collected trajectories to be used for policy evaluation, $\mathcal{D}_1$ (possibly empty), number of trajectories to collect, $n$.
2: **Output:** Data set of trajectories.
3: $k \leftarrow$ number of state-action tuples in $\mathcal{D}_1$
4: $\nabla_{\boldsymbol{\theta}}\mathcal{L} \leftarrow \frac{1}{k}\sum_{(s,a)\in\mathcal{D}_1}\nabla_{\boldsymbol{\theta}}\log\pi_{\boldsymbol{\theta}}(a|s)|_{\boldsymbol{\theta}=\boldsymbol{\theta}_e}$
5: $\mathcal{D} \leftarrow \mathcal{D}_1$
6: **for** $0 \leq i < n$ **do**
7: $\quad s_0 \sim d_0$
8: $\quad$ **for** $0 \leq t < l$ **do**
9: $\quad\quad \boldsymbol{\theta}_b \leftarrow \boldsymbol{\theta}_e - \alpha\nabla_{\boldsymbol{\theta}}\mathcal{L}$
10: $\quad\quad a_t \leftarrow A \sim \pi_{\boldsymbol{\theta}_b}(\cdot|s_t)$
11: $\quad\quad \nabla_{\boldsymbol{\theta}}\mathcal{L} \leftarrow \frac{k}{k+1}\nabla_{\boldsymbol{\theta}}\mathcal{L} + \frac{1}{k+1}\nabla_{\boldsymbol{\theta}}\log\pi_{\boldsymbol{\theta}}(a_t|s_t)|_{\boldsymbol{\theta}=\boldsymbol{\theta}_e}$
12: $\quad\quad k \leftarrow k+1$
13: $\quad\quad s_{t+1} \sim P(\cdot|s_t, a_t), r_t \leftarrow R(s_t, a_t)$
14: $\quad$ **end for**
15: $\quad \mathcal{D} \leftarrow \mathcal{D} \cup \{(s_0, a_0, r_0, ..., s_{l-1}, a_{l-1}, r_{l-1})\}$
16: **end for**
17: **Return** $\mathcal{D}$

the Monte Carlo estimate and $v(\pi_e)$ in terms of sampling error which shows how ROS's faster convergence affects the MSE of policy evaluation. These results use the following assumption:

**Assumption 1.** *The discrete state-space of the* MDP *has a directed acyclic graph (*DAG*) structure. Specifically, states in $\mathcal{S}$ can be partitioned into $l$ disjoint sets $\mathcal{S}_t$ indexed by episode step. The transition function is such that $P(s'|s,a) > 0$ implies that $s \in \mathcal{S}_t$ and $s' \in \mathcal{S}_{t+1}$.*

Note that Assumption 1 is mild as any finite-horizon MDP can be made a DAG by including the current time-step as part of the state (as we have already assumed in Section 3.1).

**Assumption 2.** ROS *uses a step-size of $\alpha \to \infty$ and the behavior policy is parameterized as a softmax function, i.e., $\pi_{\boldsymbol{\theta}}(a|s) \propto e^{\theta_{s,a}}$, where for each state, $s$, and action, $a$, we have a parameter $\theta_{s,a}$. As we formally show in Appendix B, this assumption implies that* ROS *always takes the most under-sampled action in each state.*

We also introduce the notation of $d_{\pi}^t(s)$ as the probability of visiting state $s$ at episode time $t$ while following policy $\pi$ and $d_n^t(s)$ as the empirical frequency of visitations to state $s$ at episode time $t$ after observing $n$ trajectories.

**Theorem 1.** *Under Assumptions 1 and 2 and* ROS *action selection, $d_n^t(s)$ converges to $d_{\pi}^t(s)$ with probability 1 for all $s \in \mathcal{S}$ and $0 < t < l$:*

$$\lim_{n\to\infty} d_n^t(s) = d_{\pi}^t(s), \ \forall s \in \mathcal{S}, \ 0 \leq t < l.$$

**Theorem 2.** *Let $s$ be a particular state that is visited $m$ times during data collection and assume that $|\mathcal{A}| \geq 2$. Under Assumption 2, $D_{\text{KL}}(\pi_D(\cdot|s)||\pi(\cdot|s)) = O_p(\frac{1}{m^2})$ under* ROS *sampling while $D_{\text{KL}}(\pi_D(\cdot|s)||\pi(\cdot|s)) = O_p(\frac{1}{m})$ under on-policy sampling, where $O_p$ denotes stochastic boundedness.*

**Theorem 3.** *Assume $\forall s \in \mathcal{S}, a \in \mathcal{A}$ that $R(s,a) \leq R_{\texttt{max}}$. The squared error in the Monte Carlo estimate using $\mathcal{D}$ can be upper-bounded by:*

$$\left(v(\pi_e) - \text{MC}(\mathcal{D})\right)^2 \leq \sum_{t=0}^{l-1}\gamma^{2t}R_{\texttt{max}}^2\sqrt{2D_{\text{KL}}(d_n^t||d_{\pi_e}^t) + 2\mathbf{E}_{S\sim d_n^t}[D_{\text{KL}}(\pi_D(\cdot|S)||\pi_e(\cdot|S))]}.$$

**Remark 2.** *The second term in the bound in Theorem 3 is the KL-divergence between $\pi_D$ and $\pi_e$ which Theorem 2 tells us will decrease faster under* ROS *action selection. The first term is the KL-divergence between the empirical and true state distributions which depends both on sampling*

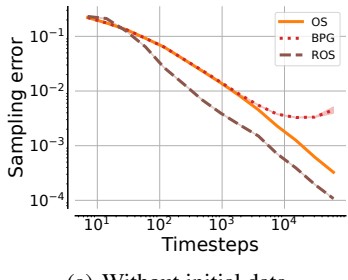
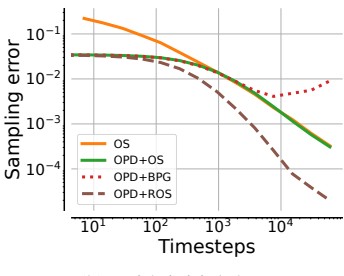

(a) Without initial data

(b) With initial data

Figure 1: Sampling error (KL) curves of data collection in the GridWorld domain. Each strategy is followed to collect data with $2^{13}\overline{T}$ steps, and all results are averaged over 200 trials with shading indicating one standard error intervals. Figures 1(a) and 1(b) show the sampling error curves of data collection **without** and **with initial data**, respectively. Axes in these figures are log-scaled.

*error in action selection as well as sampling error in the transition and initial state distributions. While the former decreases faster under ROS, the latter will decrease the same for both ROS and on-policy sampling. Hence, the theoretical faster rate of ROS for reducing sampling error in action selection may be muted by high environment stochasticity. The experimental results given in Figure 12(a) complement this theoretical observation.*

## 6 Empirical Study

We next conduct an empirical study of ROS in policy evaluation problems. Our primary goal is to answer the following questions:

1. Does ROS reduce sampling error compared to on-policy sampling?
2. Does ROS lower policy evaluation MSE when starting with and without off-policy data?

We conduct policy evaluation experiments in four domains covering discrete and continuous state and action spaces: a multi-armed bandit problem [Sutton and Barto, 1998], Gridworld [Thomas and Brunskill, 2016], CartPole, and Continuous CartPole [Brockman et al., 2016]. Since these domains are widely used, we defer their descriptions to Appendix C. Our primary baseline for comparison is on-policy sampling (OS) of i.i.d. trajectories with the Monte Carlo estimator used to compute the final policy value estimate (denoted **OS-MC**). We also compare to BPG which finds a minimum variance behavior policy for the ordinary importance sampling (OIS) policy value estimator [Hanna et al., 2017] (denoted **BPG-OIS**). We provide full experimental details concerning how $\pi_e$ and $v(\pi_e)$ were determined in Appendix C.2.

### 6.1 Policy Evaluation without Initial Data

We first run experiments in a setting **without initial data**, in which all data is collected from scratch. Letting $\overline{T}$ denote the average length of a trajectory, in each domain, we collect a total of $2^{13}\overline{T}$ environment steps with each method and compute metrics every $2^1, 2^2, ..., 2^{13}$ trajectories. Note that we specify the number of environment steps rather than number of trajectories in our empirical results. For Bandit $\overline{T} = 1$, for GridWorld $\overline{T} = 7.43$, CartPole $\overline{T} = 48.48$, and for CartPoleContinuous $\overline{T} = 49.56$. The hyper-parameter settings for all experiments are presented in Appendix E.

We first verify that ROS reduces sampling error compared to on-policy sampling. We measure sampling error with the KL-divergence (KL) between $\pi_e$ and a parametric maximum likelihood estimate of $\pi_D$ from the observe data. In Appendix D, we give a complete definition of the measure as well as an alternative measure that leads to qualitatively similar results. Due to space constraints, we only show this result for the GridWorld domain (Figure 1(a)); results for other domains are qualitatively the same and can be found in Appendix D.1. Figure 1(a) shows that with ROS sampling error decreases faster than OS. Unsurprisingly, BPG increases sampling error as it is an off-policy method which adapts the behavior policy away from $\pi_e$. These results answer our first empirical

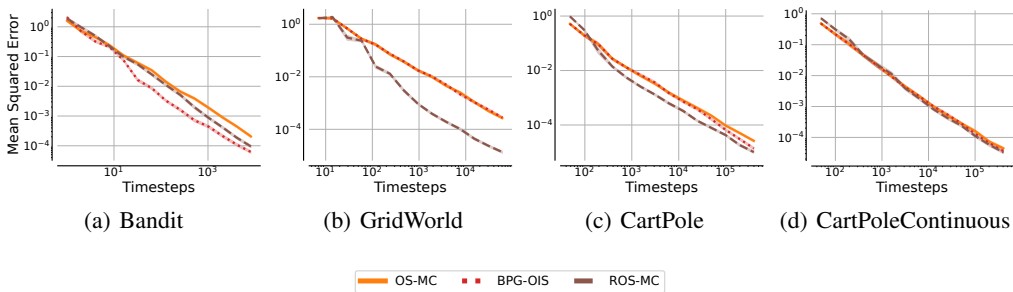

Figure 2: Mean squared error (MSE) of policy evaluation in the **without initial data** setting. Policy evaluation is conducted on the data collected from each strategy, and these curves show the MSE of the estimates (lower is better). The vertical axis gives MSE and the horizontal axis is the amount of environment steps taken (both are log-scaled). Shading indicates one standard error.

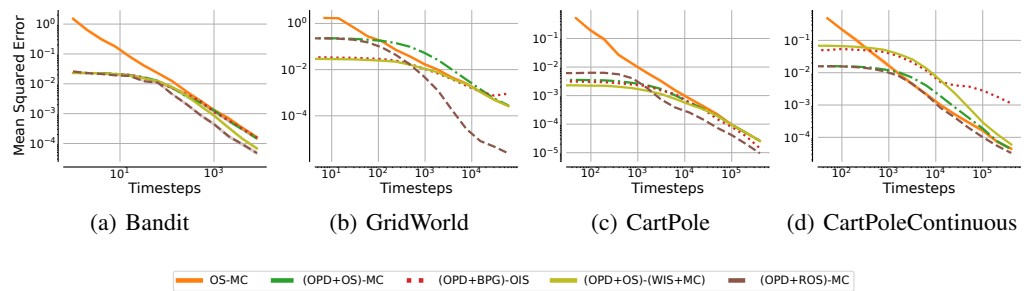

Figure 3: Mean squared error (MSE) of policy evaluation in the **with initial data** setting. Policy evaluation is conducted on the data collected from each strategy and a small set of initial data collected off-policy. Axes and confidence intervals are the same as in Figure 2.

question and confirm our theoretical claim that non-i.i.d. off-policy sampling can cause the empirical distribution of data to converge to the expected on-policy distribution faster.

Ultimately, this paper focuses on reducing sampling error for lower MSE policy evaluation. Figure 2 shows that ROS lowers MSE compared to both OS and BPG across all domains.[2] These results address our second empirical question and support the claim that reducing sampling error decreases the MSE of the Monte Carlo estimator for policy evaluation.

## 6.2 Policy Evaluation with Initial Data

Our next set of experiments considers a setting **with initial data**, in which a set of 100 trajectories are already available and we wish to use these trajectories in our policy value estimate. These trajectories are collected via i.i.d. *off-policy* sampling with a behavior policy that is slightly different than $\pi_e$. This setting is intended to represent a setting where $\pi_e$ has just been updated from an older policy and we would like to still use the off-policy data already collected from the older policy combined with the data to be collected. In addition to the off-policy data (OPD), we collect an additional $2^{13}\overline{T}$ steps of environment interaction with each method. *We do not count the initial 100 trajectories towards the total collected data.*

For ROS, we use the OPD to initialize $\nabla_{\boldsymbol{\theta}}\mathcal{L}$. We expect to see that ROS will collect data to combine with the OPD such that the aggregate data set looks as if it had been collected with $\pi_e$ to begin with. We compare ROS to the following baseline methods. **(OPD + OS)-MC** collects additional data with OS and uses the Monte Carlo estimator with the total data-set. **(OPD + OS)-(WIS + MC)** uses weighted importance sampling (WIS) to compute an estimate from the OPD combines the WIS estimate with a Monte Carlo estimate using on-policy data. **(OPD + BPG)-OIS** collects additional data with BPG and uses ordinary importance sampling as the estimator with all data. Finally, **(OS - MC)** replaces the 100

---
[2]Numeric values for the final MSE of each method can be found in Appendix F. We also report median and interquartile ranges of the error of each method in Appendix G.

initial trajectories with trajectories from OS, then collects the remaining data with OS and uses the Monte Carlo estimator. In the case of **(OS - MC)**, the 100 initial trajectories are counted towards the total data collected.

Figure 1(b) shows that sampling error decreases fastest for ROS as the additional data is collected. For policy evaluation, we show MSE for varying amounts of data in Figure 3 and provide numerical values for the final MSE in Appendix F. We observe that the initial data provides an immediate reduction in MSE at the expense of injecting bias into the estimates. **(OPD+OS)-MC** struggles to reduce this bias while **(OPD+ROS)-MC** is able to through data collection. Overall, this result highlights that ROS can collect additional data that reduces sampling error in the aggregate data set and produce lower MSE estimates compared to other data collection methods. Intuitively, OS requires many more samples to dilute the bias brought on by using OPD in the Monte Carlo estimator, while ROS is able to correct the empirical off-policy distribution to the expected on-policy distribution and use the Monte Carlo estimator without any off-policy corrections.

The comparison to OS-MC demonstrates the potential of ROS for correcting an off-policy empirical distribution to the expected on-policy distribution. As noted above, OS-MC has 100 fewer trajectories than the other baselines. However – even when including the initial 100 off-policy trajectories in the data total for all methods – ROS eventually obtains lower MSE compared to OS-MC. In this sense, ROS has taken an initially biased dataset and collected the right trajectories to make it look as-if the evaluation policy had collected all trajectories in the first place.

### 6.3 Sensitivity Study

Finally, we evaluate the sensitivity of ROS to hyper-parameter, environment, and policy settings. ROS requires setting a step size, $\alpha$, which controls how much ROS updates the behavior policy away from $\pi_e$. We show MSE curves for ROS with different step size $\alpha$ on GridWorld and CartPole in Figures 4 ($\alpha = 0$ corresponds to OS). Figure 4(a) shows that, in GridWorld, ROS with any tested step-size produces lower MSE policy evaluation than OS for any data set size. As it collects more data, ROS with larger $\alpha$ enables lower MSE because the norm of $\nabla_{\boldsymbol{\theta}}\mathcal{L}$ decreases as sampling error decreases, and thus a larger $\alpha$ is required to make significant updates. A larger $\alpha$ value is also in line with our theoretical results which prescribe $\alpha \to \infty$. However, in CartPole, (Figure 4(b)), ROS with the largest tested $\alpha$ (1000) diverges and the second largest ($\alpha = 100$) requires many steps before it improves upon OS. Thus, in domains with continuous state-spaces, more conservative $\alpha$ values may be preferred.

Our final set of experiments considers how the stochasticity of a domain and entropy of $\pi_e$ affect the relative improvement that ROS offers. In this sub-section, we study these settings in the Bandit domain for its simplicty; similar experimental results in GridWorld can be found in Appendix H. We choose $\alpha = 1000$ for the following experiments.

To study domain stochasticity, we first create variants of the Bandit environment by multiplying either the mean or scale of the reward distribution of each action by a varying factor. In each experimental trial, we use ROS to collect $1000\overline{T}$ steps for the Monte Carlo estimator and compute the relative MSE compared to the Monte Carlo estimator using OS with the same number of steps. Figure 4(c) shows that as the factor on the mean increases, ROS provides a greater reduction in MSE as even small amounts of sampling error translate into large MSE when the reward means are large. On the other hand, as the scale factor increases, the MSE is dominated by reward noise and the relative benefit of reducing sampling error disappears.

We also evaluate the relative improvement of ROS as a function of the entropy of $\pi_e$. For $\pi_e$, we use $\epsilon$-greedy policies which select the optimal action in a state with probability $1 - \epsilon$ and otherwise select an action uniformly at random. Relative improvement in MSE is shown in Figure 4(d). For all $\epsilon$, ROS improves upon the MSE of OS. The improvement is generally larger for more stochastic $\pi_e$ when sampling error in action selection will be highest.

## 7  Discussion and Future Work

This work has shown that off-policy non-i.i.d. sampling can produce data sets that more closely approximate the on-policy data distribution than on-policy i.i.d. sampling. We considered the problem of policy evaluation and showed that more closely approximating the on-policy data distribution leads to more data efficient policy evaluation across several domains. As far as we know, ROS is the first

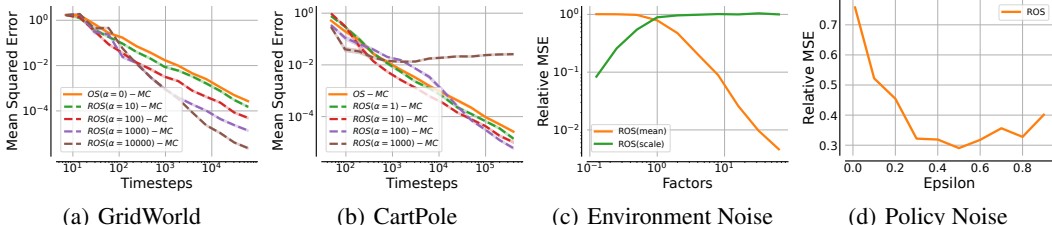

| (a) GridWorld | (b) CartPole | (c) Environment Noise | (d) Policy Noise |

Figure 4: MSE of ROS with different step-size, $\alpha$ (4(a) and 4(b)). Relative improvement of ROS in Bandit compared to OS with different stochasticity in the environment (4(c)) and policy (4(d)). The relative MSE is computed as the MSE of ROS divided by the MSE of OS. Results in these figures are averaged over 500 trials.

data collection method for policy evaluation that uses off-policy sampling to produce more closely on-policy data than the data produced by on-policy sampling.

While ROS is a first step towards off-policy algorithms that produce data matching a target distribution, we highlight a few limitations of the algorithm and our study. In our view, the main limitations of the ROS algorithm are the need to set a step-size parameter (in contrast to parameter-free on-policy sampling) and the need to update $\nabla_{\boldsymbol{\theta}} \mathcal{L}$ at each action step. For the former, future work should investigate robust methods for setting the step-size, particularly in settings where $\pi_{\boldsymbol{\theta}}$ generalizes across the state-space. For the latter limitation, a future study could consider only updating $\nabla_{\boldsymbol{\theta}} \mathcal{L}$ at the end of each episode instead of after each action choice (assuming more computation can be done between episodes). In terms of our study, for this paper we chose to study many different facets of ROS on a suite of simpler domains (see the appendices for additional ablations and extensions); a future study should assess the scalability of ROS with more complex function approximators. Finally, our theoretical results were conducted in the tabular setting; an important open question is at what rate ROS converges when $\pi_{\boldsymbol{\theta}}$ uses a function approximator that must generalize across states. Beyond these minor technical limitations, our paper addresses fundamental research questions in RL and thus we do not see obvious negative societal impacts that are unique to this work in comparison to other work in RL and policy evaluation.

While we evaluated ROS for policy evaluation, the long-term importance of this work may be in exploring the distinction between on-policy sampling and on-policy data. On-policy RL algorithms require on-policy data and our work suggests that adaptive off-policy sampling can produce on-policy data more efficiently than on-policy sampling. In the future, we wish to study these insights for on-policy policy improvement algorithms (e.g., policy gradient methods [Williams, 1992, Schulman et al., 2017]) and to extend our convergence results to non-tabular settings.

## 8 Conclusion

In this paper, we have introduced a novel data collection method for policy evaluation in reinforcement learning environments. Our method – Robust On-Policy Sampling (ROS) – considers previously collected data when selecting actions to reduce sampling error in the entire collected data set. We show both in theory and in practice that data from ROS converges faster to the on-policy data distribution compared to on-policy sampling. Empirically, we find that faster convergence to the on-policy data distributions lowers the MSE of policy evaluation.

## Acknowledgments and Disclosure of Funding

We thank Ishan Durugkar, Brahma Pavse, Subhojyoti Mukherjee, Elliot Fosong, and Filippos Christianos for their feedback which greatly strengthened the paper. We also wish to acknowledge the anonymous reviewers for their comments and constructive criticisms. Support for this research was provided by the Office of the Vice Chancellor for Research and Graduate Education at the University of Wisconsin — Madison with funding from the Wisconsin Alumni Research Foundation.

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
