# A  Proof of Proposition 1

In this appendix we prove Proposition 1 from Section 4.

**Proposition 1.** *The data conditioned Monte Carlo estimator is biased under on-policy sampling of $D_2$ unless $\mathrm{MC}(\mathcal{D}_1) = v(\pi_e)$ or $\mathcal{D}_1 = \emptyset$. That is:*

$$\mathbf{E}\left[\mathrm{MC}(\mathcal{D}_1 \cup D_2) \,\middle|\, D_2 \sim \pi_e\right] \neq v(\pi_e).$$

*Proof.* In the following, we use $\mathbf{E}_{\pi_e}[\cdot]$ as shorthand for $\mathbf{E}[\cdot|D \sim \pi_e]$. Let $\hat{v}_1 := \frac{1}{n_{\mathcal{D}_1}} \sum_{i=1}^{n_{\mathcal{D}_1}} g(h_i) = \mathrm{MC}(\mathcal{D}_1)$.

$$\mathbf{E}_{\pi_e}\left[\mathrm{MC}(\mathcal{D}_1 \cup D_2)\right] = \mathbf{E}_{\pi_e}\left[\frac{1}{n}\sum_{i=1}^{n_{\mathcal{D}_1}} g(h_i) + \frac{1}{n}\sum_{i=1}^{n_{D_2}} g(H_i)\right] \tag{5}$$

$$= \mathbf{E}_{\pi_e}\left[\frac{1}{n}\sum_{i=1}^{n_{\mathcal{D}_1}} g(h_i)\right] + \mathbf{E}_{\pi_e}\left[\frac{1}{n}\sum_{i=1}^{n_{D_2}} g(H_i)\right] \tag{6}$$

$$\stackrel{(a)}{=} \frac{1}{n}\frac{n_{\mathcal{D}_1}}{n_{\mathcal{D}_1}}\sum_{i=1}^{n_{\mathcal{D}_1}} g(h_i) + \mathbf{E}_{\pi_e}\left[\frac{1}{n}\frac{n_{D_2}}{n_{D_2}}\sum_{i=1}^{n_{D_2}} g(H_i)\right] \tag{7}$$

$$\stackrel{(b)}{=} \frac{n_{\mathcal{D}_1}}{n}\hat{v}_1 + \frac{n_{D_2}}{n}\mathbf{E}_{\pi_e}\left[\underbrace{\frac{1}{n_{D_2}}\sum_{i=1}^{n_{D_2}} g(H_i)}_{\mathrm{MC}(D_2)}\right] \tag{8}$$

$$\stackrel{(c)}{=} \frac{n_{\mathcal{D}_1}}{n}\hat{v}_1 + \frac{n_{D_2}}{n}v(\pi_e) \tag{9}$$

$$= \frac{n_{\mathcal{D}_1}}{n}(\hat{v}_1 - v(\pi_e)) + \frac{n_{D_2}}{n}v(\pi_e) + \frac{n_{\mathcal{D}_1}}{n}v(\pi_e) \tag{10}$$

$$= \frac{n_{\mathcal{D}_1}}{n}(\hat{v}_1 - v(\pi_e)) + v(\pi_e) \tag{11}$$

where **(a)** uses the fact that no random variables appear inside the first expectation, **(b)** uses the definition of $\hat{v}$, and **(c)** uses the fact that $\mathrm{MC}(D_2)$ is an unbiased estimator of $v(\pi_e)$. The proposition follows by observing that equation 11 is equal to $v(\pi_e)$ if and only if $n_{\mathcal{D}_1} = 0$ (i.e., $\mathcal{D}_1 = \emptyset$) or if $\hat{v} = v(\pi_e)$.

Note that the latter case in which the data-conditioned Monte Carlo estimator is unbiased only occurs when the Monte Carlo estimate *only using* $\mathcal{D}_1$ has non-zero error. If $g(H)$ has non-zero variance under on-policy sampling then the probability that $\mathrm{MC}(\mathcal{D}_1)$ is exactly equal to $v(\pi_e)$ with a finite-size $D_2$ is low.

$\square$

# B  Proofs of ROS Properties

Before giving the proofs for Theorems 1 and 2 we re-state our key assumption.

**Assumption 2.** ROS *uses a step-size of $\alpha \to \infty$ and the behavior policy is parameterized as a softmax function, i.e., $\pi_{\boldsymbol{\theta}}(a|s) \propto e^{\theta_{s,a}}$, where for each state, $s$, and action, $a$, we have a parameter $\theta_{s,a}$. As we formally show in Appendix B, this assumption implies that ROS always takes the most under-sampled action in each state.*

We next derive two lemmas that will be used in the proofs of our theorems.

**Lemma 1.** *Let $S_m(a)$ denote the number of times that action $a$ was taken after visiting a particular state, $s$, $m$ times in $\mathcal{D}$ and let $k := |\mathcal{A}|$. Under Assumption 2 and ROS collection of $\mathcal{D}$, we have that:*

$$\sup_{a \in \mathcal{A}} |S_m(a) - m\pi_e(a|s)| \leq k - 1. \tag{12}$$

*Proof.* We first formally show that Assumption 2 implies that ROS always taking the most under-sampled action at each time-step. Because we only consider a single state, we suppress dependencies on the state in this proof, e.g., we write $\pi_e(a)$ instead of $\pi_e(a|s)$ and $\theta_a$ instead of $\theta_{s,a}$ for the softmax parameters. We have $\mathcal{L}(\pi_{\boldsymbol{\theta}}) = \sum_a S_m(a)\theta_a - m\log(\sum_{b\in\mathcal{A}} e^{\theta_b})$ and $\nabla_{\theta_a}\mathcal{L}(\pi_{\boldsymbol{\theta}})|_{\pi_{\boldsymbol{\theta}}=\pi_e} = S_m(a)(1-\pi_e(a)) - \pi_e(a)(m - S_m(a)) = S_m(a) - m\pi_e(a)$, which is exactly the number of times that action $a$ was over-sampled. The softmax parameter for action $a$ after the ROS update is given by:

$$\theta'_a = \theta_{e,a} - \alpha(S_m(a) - m\pi_e(a))$$

where $\theta_{e,a}$ is the softmax parameter for action $a$ under the evaluation policy. Taking the limit as $\alpha \to \infty$, we see that the $\theta_{e,a}$ term is dominated by the $\alpha(S_m(a) - \pi_e(a)m)$ term. Thus the parameter for each action is:

$$\theta'_a = \alpha((m\pi_e(a) - S_m(a)).$$

Interpreting $\alpha$ as the inverse of the softmax temperature, we can see that $\alpha \to \infty$ corresponds to a softmax temperature $\to 0$ which in turn corresponds to a hard max over the $(m\pi_e(a) - S_m(a))$ values. Thus, the updated behavior policy puts all probability mass on the action for which $\pi_e(a)m - S_m(a)$ is largest. Hence we select the most under-sampled action if we take $\alpha \to \infty$ in Algorithm 1.

Now we show that under ROS action selection that the amount of over- or under-sampling is bounded. First, we observe that,

$$\sum_{a\in\mathcal{A}}(S_m(i) - m\pi_e(i)) = m - m = 0,$$

and we claim that,

$$S_m(a) - m\pi_e(a) \leq 1$$

for any $a \in \mathcal{A}$. We prove this claim by contradiction. Assume not: $S_m(j) - m\pi_e(j) > 1$ for some $j \in \mathcal{A}$. Let $X_m(j) := 1$ if action $j$ was taken at step $m$ and 0 otherwise. If $X_m(j) = 1$, we have that:

$$S_m(j) - m\pi_e(i) > 1 \implies$$
$$S_{m-1}(j) + 1 - (m-1)\pi_e(j) - \pi_e(j) > 1 \implies$$
$$S_{m-1}(j) - (m-1)\pi_e(j) > 1 - 1 + \pi_e(j) > 0.$$

However, this results in a contradiction since we have that action $j$ was over-sampled at the previous step but action $j$ would not be selected by ROS if it was over-sampled. So, in order for $S_m(j) - m\pi_e(j) > 1$, we must have that $X_m(j) = 0$. Combining the fact that $X_m(j) = 0$ with our assumption that $S_m(j) - m\pi_e(j) > 1$ tells us that:

$$S_m(j) - m\pi_e(j) > 1$$
$$\overset{(a)}{\implies} S_{m-1}(j) - m\pi_e(j) > 1$$
$$\implies S_{m-1}(j) - (m-1)\pi_e(j) > 1$$

where **(a)** is because $S_{m-1}(j)$ must be equal to $S_m(j)$ if $X_m(j) = 0$. By the same logic we get that $X_{m-1}(j) = x_{m-2}(j) = \cdots = X_1(j) = 0$ which implies that $S_1(j) - \pi_e(j) = 0 - \pi_e > 1$ which is a contradiction. Thus, we conclude that $S_m(j) - m\pi_e(j) \leq 1$ for any $j$, i.e., any action can be *over-sampled* by at most 1. Combining this conclusion with the observation that $\sum_{a\in\mathcal{A}} S_m(a) - m\pi_e(a) = m - m = 0$ tells us that any action can be *under-sampled* by at most $k - 1$. Thus, the absolute difference $|S_m(a) - m\pi_e(a)|$ is at most $k - 1$ for any action $a \in \mathcal{A}$ which completes the proof.

$\square$

**Lemma 2.** *Let $s$ be a state that we visit $m$ times. Under ROS sampling, we have $\forall a \in \mathcal{A}$ that:*

$$\lim_{m\to\infty} \pi_D(a|s) = \pi_e(a|s).$$

*Proof.* The proof follows from Lemma 1. As in the proof of Lemma 1, we suppress the dependency on $s$ in our notation since we are only concerned with a fixed state. Using the notation from the proof

of Lemma 1, we have that $\pi_D(a) = \frac{S_m(a)}{m}$.

$$|S_m(a) - m\pi_e(a)| \leq k - 1$$
$$\implies \frac{|S_m(a) - m\pi_e(a)|}{m} \leq \frac{k-1}{m}$$
$$\implies \lim_{m\to\infty} \frac{|S_m(a) - m\pi_e(a)|}{m} \leq \lim_{m\to\infty} \frac{k-1}{m}$$
$$\implies \lim_{m\to\infty} |\pi_D(a) - \pi_e(a)| \leq 0$$
$$\implies \lim_{m\to\infty} |\pi_D(a) - \pi_e(a)| = 0$$
$$\implies \lim_{m\to\infty} \pi_D(a) = \pi_e(a).$$

$\square$

**Theorem 1.** *Under Assumptions 1 and 2 and* ROS *action selection, $d_n^t(s)$ converges to $d_\pi^t(s)$ with probability 1 for all $s \in \mathcal{S}$ and $0 < t < l$:*

$$\lim_{n\to\infty} d_n^t(s) = d_\pi^t(s), \ \forall s \in \mathcal{S}, \ 0 \leq t < l.$$

*Proof.* The proof is by induction. For the base case, note that $d_\pi^0(s) = d_0(s)$ and thus $\lim_{n\to\infty} d_n^0(s) = d_\pi^0(s) \ \forall s$ with probability 1 by the strong law of large numbers.

For the induction step, we assume $\lim_{n\to\infty} d_n^t(s) = d_\pi^t(s) \ \forall s$ with probability 1 for some episode step $t < l$. We want to show that this implies that $\lim_{n\to\infty} d_n^{t+1}(s) = d_\pi^{t+1}(s)$ with probability 1 for all $s$. Let $P_n$ denote the empirical state transition function, i.e., $P_n(s'|s,a) := \frac{c_n(s,a,s')}{c_n(s,a)}$ where $c_n(s,a,s')$ is the number of times that $(s,a,s')$ occurred in $D$ and similarly for $c_n(s,a)$. Note that $d_n^{t+1}(s) = \sum_{\tilde{s}} \sum_a d_n^t(\tilde{s})\pi_D(a|\tilde{s})P_n(s|\tilde{s},a)$ and $d_\pi^{t+1}(s) = \sum_{\tilde{s}} \sum_a d_\pi^t(\tilde{s})\pi(a|\tilde{s})P(s|\tilde{s},a)$. The former claim follows as a consequence of the finite-horizon MDP setting in which the state implicitly must depend on the current time-step. In this setting $P_n(s'|s,a)$ and $\pi_D(a|s)$ are only computed with samples from a particular time-step (or pair of subsequent time-steps in the case of $P_n$). Then we have that:

$$\lim_{n\to\infty} d_n^{t+1}(s) = \lim_{n\to\infty} \sum_{\tilde{s}} \sum_a d_n^t(\tilde{s})\pi_D(a|\tilde{s})P_n(s|\tilde{s},a)$$
$$= \sum_{\tilde{s}} \sum_a \lim_{n\to\infty} d_n^t(\tilde{s})\pi_D(a|\tilde{s})P_n(s|\tilde{s},a)$$
$$= \sum_{\tilde{s}} \sum_a \lim_{n\to\infty} d_n^t(\tilde{s}) \cdot \lim_{n\to\infty} \pi_D(a|\tilde{s}) \cdot \lim_{n\to\infty} P_n(s|\tilde{s},a)$$
$$= \sum_{\tilde{s}} \sum_a \lim_{n\to\infty} d_n^t(\tilde{s}) \cdot \lim_{n\to\infty} \pi_D(a|\tilde{s}) \cdot \lim_{n\to\infty} P_n(s|\tilde{s},a)$$
$$\overset{(a)}{=} \sum_{\tilde{s}} \sum_a d_\pi^t(\tilde{s})\pi(a|\tilde{s})P(s|\tilde{s},a)$$
$$= d_\pi^{t+1}(s)$$

where **(a)** follows from the induction hypothesis, Lemma 2, and $\lim_{n\to\infty} P_n(s'|s,a) = P(s'|s,a)$ follows from the strong law of large numbers. This completes the proof.

$\square$

**Theorem 2.** *Let $s$ be a particular state that is visited $m$ times during data collection and assume that $|\mathcal{A}| \geq 2$. Under Assumption 2, $D_{\text{KL}}(\pi_D(\cdot|s)||\pi(\cdot|s)) = O_p(\frac{1}{m^2})$ under* ROS *sampling while $D_{\text{KL}}(\pi_D(\cdot|s)||\pi(\cdot|s)) = O_p(\frac{1}{m})$ under on-policy sampling, where $O_p$ denotes stochastic boundedness.*

*Proof.* This proof has two parts. Since we only consider a particular state, we suppress the state-dependency in policies throughout this proof, i.e., we write $\pi(a)$ instead of $\pi(a|s)$. The on-policy

sampling rate is adapted from Theorem 1 of Mardia et al. [2019] which implies that, the KL-divergence between the empirical distribution of a discrete distribution and that discrete distribution itself is $O_p(\frac{1}{m})$. In our case, $\pi_e$ is the discrete distribution and $\pi_D$ is the empirical distribution. Then we have from Mardia et al. [2019] that $2mD_{\text{KL}}(\pi_D||\pi_e) \xrightarrow{d} \chi^2_{k-1}$ where $k$ is the number of actions. So we have that $D_{KL}(\pi_D||\pi_e) = O_p(\frac{1}{m})$ from Lemma 5.3 of Alexanderian [2009] since convergence in distribution implies bounded in probability.

Second, we obtain the ROS rate. Define the vectors $\tilde{\pi}_e := (\pi_e(1), \cdots, \pi_e(k-1))$ and $\tilde{\pi}_D := (\pi_D(1), \cdots, \pi_D(k-1))$. We write the KL-divergence between $\pi_e$ and $\pi_D$ as a function of these two vectors:

$$D_{KL}(\tilde{\pi}_D||\tilde{\pi}_e) = \sum_{i=1}^{k-1} \tilde{\pi}_D(i) \log(\frac{\tilde{\pi}_D(i)}{\tilde{\pi}_e(i)}) + (1 - \sum_{j=1}^{k-1} \tilde{\pi}_D(j)) \log(\frac{1 - \sum_{j=1}^{k-1} \tilde{\pi}_D(j)}{1 - \sum_{j=1}^{k-1} \tilde{\pi}_e(j)}).$$

Let $\mathbf{g}$ denote the gradient of $D_{KL}(\tilde{\pi}_D||\tilde{\pi}_e)$ with respect to $\tilde{\pi}_D$, evaluated at point $\tilde{\pi}_e$ and $\mathbf{H}$ denotes the Hessian matrix of $D_{KL}(\pi_D||\pi_e)$ evaluated at point $\tilde{\pi}_e$. Clearly, $\mathbf{g} = \mathbf{0}$ because setting $\pi_D$ to $\pi_e$ will minimize $D_{\text{KL}}(\pi_D||\pi_e)$. We take the Hessian from Mardia et al. [2019]:

$$\mathbf{H}(i,j) = \begin{cases} \frac{1}{\pi_e(k)} & i \neq j \\ \frac{1}{\pi_e(i)} + \frac{1}{\pi_e(k)} & i = j \end{cases}$$

Alternatively, we can write $\mathbf{H} = \frac{1}{\pi_e(k)} \mathbf{1}\mathbf{1}^\top + \texttt{diag}(\tilde{\pi}_e^{-1})$ where $\texttt{diag}(\tilde{\pi}_e^{-1})$ is the $(k - 1 \times k - 1)$ matrix with $\frac{1}{\tilde{\pi}_e}$ on its diagonal. Using a Taylor Series expansion for $D_{\text{KL}}$, we have:

$$m^2 D_{\text{KL}}(\pi_D||\pi_e) = m^2 D_{\text{KL}}(\pi_e||\pi_e) + m^2 \mathbf{g}^T(\tilde{\pi}_D - \tilde{\pi}_e) + \underbrace{\frac{1}{2} m(\tilde{\pi}_D - \tilde{\pi}_e)^T \mathbf{H} m(\tilde{\pi}_D - \tilde{\pi}_e)}_{\text{Quadratic Term}} + Q_m$$

where $Q_m$ consists of the higher order terms. Note that since $D_{\text{KL}}(\pi_e||\pi_e) = 0$, the first two terms are both 0 so we only need to bound the quadratic term and higher terms. By Lemma 1 every component of $m(\tilde{\pi}_D - \tilde{\pi}_e)$ is bounded by $k - 1$, so the quadratic term is bounded by $\frac{1}{2} \sum_{i,j} \mathbf{H}(i,j)(k-1)^2$, which is of order $O(1)$ since $\mathbf{H}$ is a constant when evaluated at $\tilde{\pi}_e$. The higher order terms $Q_m$ contain higher powers of $(\tilde{\pi}_D - \tilde{\pi}_e)$ multiplied by $m^2$. Using Lemma 1, we can bound all components of any $m(\tilde{\pi}_D - \tilde{\pi}_e)$ by $k - 1$ to replace each $m$ with a constant (trivially, all components of $(\tilde{\pi}_D - \tilde{\pi}_e)$ can also be bounded by $k - 1$) and higher-order derivatives of $D_{\text{KL}}$ evaluated at $\pi_e$ are also constant with respect to $m$. Thus higher order terms are also $O(1)$ which gives us that $m^2 D_{\text{KL}}(\pi_D||\pi_e) = O(1)$ and thus that $D_{\text{KL}}(\pi_D||\pi_e) = O(\frac{1}{m^2})$. Deterministic convergence to zero implies probabilistic convergence and thus, under ROS action selection, $D_{\text{KL}}(\pi_D||\pi_e) = O_p(\frac{1}{m^2})$. ☐

**Theorem 3.** *Assume $\forall s \in \mathcal{S}, a \in \mathcal{A}$ that $R(s,a) \leq R_{\texttt{max}}$. The squared error in the Monte Carlo estimate using $\mathcal{D}$ can be upper-bounded by:*

$$(v(\pi_e) - \text{MC}(\mathcal{D}))^2 \leq \sum_{t=0}^{l-1} \gamma^{2t} R_{\texttt{max}}^2 \sqrt{2D_{\text{KL}}(d_n^t||d_{\pi_e}^t) + 2\mathbf{E}_{S \sim d_n^t}[D_{\text{KL}}(\pi_D(\cdot|S)||\pi_e(\cdot|S)]}.$$

*Proof.* First, we introduce some additional notation. We define $\mu_n^t(s,a) := \frac{n_t(s,a)}{n}$ as the empirical distribution of a state-action pair at time $t$, where $n_t(s,a)$ is the number of times that state $s$ and action $a$ occur jointly at time $t$ across trajectories in $\mathcal{D}$. Let $\mu_{\pi_e}^t(s,a)$ be the probability of state $s$ and action $a$ occurring jointly at time $t$ while following $\pi_e$. Note that $\mu_n^t(s,a) = d_n^t(s)\pi_D(a|s)$ and $\mu_{\pi_e}^t(s,a) = d_{\pi_e}^t(s)\pi_e(a|s)$. Finally, we define $n(s_0, a_0, \ldots, s_{l-1}, a_{l-1})$ as the number of times that state-action trajectory $s_0, a_0, \ldots, s_{l-1}, a_{l-1}$ occurs in $\mathcal{D}$.

Observe that the Monte Carlo estimate can be re-written as:

$$
\begin{aligned}
\mathrm{MC}(D) =& \frac{1}{n}\sum_{i=1}^{n}\sum_{t=0}^{l-1}\gamma^{t}R(s_{i,t},a_{i,t})\\
=& \frac{1}{n}\sum_{s_0}\sum_{a_0}\cdots\sum_{s_{l-1}}\sum_{a_{l-1}}n(s_0,a_0,\ldots,s_{l-1},a_{l-1})\sum_{t=0}^{l-1}\gamma^{t}R(s_t,a_t)\\
=& \frac{1}{n}\sum_{t=0}^{l-1}\gamma^{t}\sum_{s_0}\sum_{a_0}\cdots\sum_{s_{l-1}}\sum_{a_{l-1}}n(s_0,a_0,\ldots,s_{l-1},a_{l-1})R(s_t,a_t)\\
=& \frac{1}{n}\sum_{t=0}^{l-1}\gamma^{t}\sum_{s}\sum_{a}n_t(s,a)R(s,a)\\
=& \sum_{t=0}^{l-1}\gamma^{t}\sum_{s}\sum_{a}\mu_n^t(s,a)R(s,a).
\end{aligned}
$$

Similarly, the true value of the evaluation policy can be re-written in terms of the state-action distribution at each time step under policy $\pi_e$:

$$
\begin{aligned}
v(\pi_e) =& \sum_{h}\Pr(h|\pi_e)\sum_{t=0}^{l-1}\gamma^{t}R(s_{h,t},a_{h,t})\\
=& \sum_{s_0}\sum_{a_0}\cdots\sum_{s_{l-1}}\sum_{a_{l-1}}\underbrace{d_0(s_0)\pi_e(a_0|s_0)\prod_{i=1}^{l-1}\pi_e(a_i|s_i)P(s_i|s_{i-1},a_{i-1})}_{=\Pr(h=(s_0,a_0,\ldots,s_{l-1},a_{l-1})|\pi_e)}\sum_{t=0}^{l-1}\gamma^{t}R(s_t,a_t)\\
=& \sum_{t=0}^{l-1}\gamma^{t}\sum_{s_0}\sum_{a_0}\cdots\sum_{s_{l-1}}\sum_{a_{l-1}}d_0(s_0)\pi_e(a_0|s_0)\cdots P(s_{l-1}|s_{l-2},a_{l-2})R(s_t,a_t)\\
=& \sum_{t=0}^{l-1}\gamma^{t}\sum_{s}\sum_{a}\mu_{\pi_e}^t(s,a)R(s,a)
\end{aligned}
$$

Now, we use these alternative formulations to bound the squared error between the true value and the Monte Carlo estimate computed with a fixed data-set $D$.

$$
\begin{aligned}
(v(\pi_e)-\mathrm{MC}(D))^2 =& \left(\sum_{t=0}^{l-1}\gamma^{t}\sum_{s}\sum_{a}R(s,a)(\mu_n^t(s,a)-\mu_{\pi_e}^t(s,a))\right)^2\\
\overset{(a)}{\leq}& \sum_{t=0}^{l-1}\gamma^{2t}\sum_{s}\sum_{a}\left(R(s,a)(\mu_n^t(s,a)-\mu_{\pi_e}^t(s,a))\right)^2\\
\overset{(b)}{\leq}& \sum_{t=0}^{l-1}\gamma^{2t}R_{\max}^2\sum_{s}\sum_{a}\left(\mu_n^t(s,a)-\mu_{\pi_e}^t(s,a)\right)^2\\
\overset{(c)}{\leq}& \sum_{t=0}^{l-1}\gamma^{2t}R_{\max}^2\sum_{s}\sum_{a}\left|\mu_n^t(s,a)-\mu_{\pi_e}^t(s,a)\right|\\
\overset{(d)}{\leq}& \sum_{t=0}^{l-1}\gamma^{2t}R_{\max}^2\sqrt{2D_{\mathrm{KL}}(\mu_n^t||\mu_{\pi_e}^t)}
\end{aligned}
$$

where (a) uses Jensen's inequality, (b) replaces $R(s,a)$ with the constant $r_{\max}$, (c) notes that $-1\geq\mu_n^t(s,a)-\mu_{\pi_e}^t(s,a)\leq 1$ so $|\mu_n^t(s,a)-\mu_{\pi_e}^t(s,a)|\geq(\mu_n^t(s,a)-\mu_{\pi_e}^t(s,a))^2$, and (d) uses Pinsker's inequality. All that remains is to use the definition of the KL-divergence and properties of logarithms

and expectations to obtain the final form of the bound:

$$\sum_{t=0}^{l-1} \gamma^{2t} R_{\max}^2 \sqrt{2D_{\text{KL}}(\mu_n^t || \mu_{\pi_e}^t)} = \sum_{t=0}^{l-1} \gamma^{2t} R_{\max}^2 \sqrt{2E_{S \sim d_n^t, A \sim \pi_D}[\log \frac{d_n^t(S)\pi_D(A|S)}{d_{\pi_e}^t(S)\pi_e(A|S)}]}$$

$$= \sum_{t=0}^{l-1} \gamma^{2t} R_{\max}^2 \sqrt{2E_{S \sim d_n^t, A \sim \pi_D}[\log \frac{d_n^t(S)}{d_{\pi_e}^t(S)} + \log \frac{\pi_D(A|S)}{\pi_e(A|S)}]}$$

$$= \sum_{t=0}^{l-1} \gamma^{2t} R_{\max}^2 \sqrt{2D_{\text{KL}}(d_n^t || d_{\pi_e}^t) + 2\mathbf{E}_{S \sim d_n^t}[D_{\text{KL}}(\pi_D(\cdot|S)||\pi_e(\cdot|S))]}.$$

$\square$

## C   Experiment Domains

This appendix provides additional details on our experimental set-up.

### C.1   Extended Domain Descriptions

This section describes the domains used in our empirical evaluation. Figure 5 illustrates each domain.

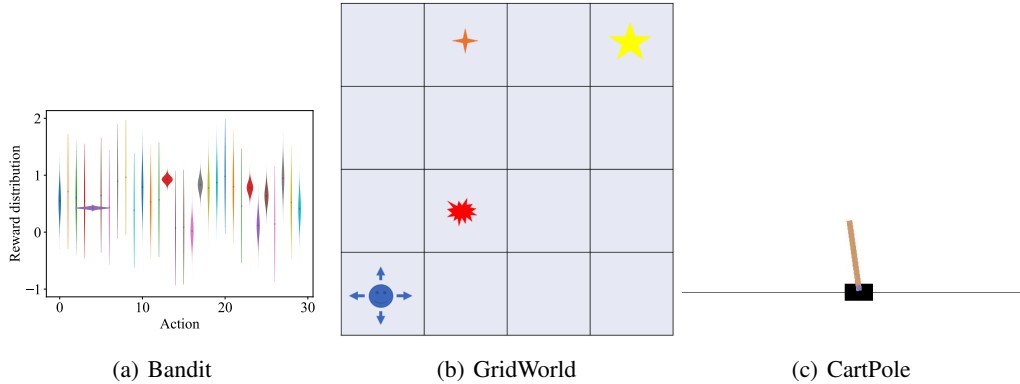

(a) Bandit       (b) GridWorld       (c) CartPole

Figure 5: Experimental Domains. Figure 5(a) shows the distribution of rewards from each action in a 30-armed bandit problem, with black dots indicating the mean rewards and shading indicating the scale of the distribution. Figure 5(b) shows a $4 \times 4$ GridWorld with the agent starting from the bottom-left corner and the goal state in the top-right corner. Figure 5(c) shows the CartPole environment in which the goal is to keep the pole from falling over. ContinuousCartPole is the same as CartPole except with a modified action space.

Our first domain, Bandit, is a 30-armed bandit problem modelled after the 10-armed bandit in [Sutton and Barto [1998, Chapter 2]]. This domain has a single state, 30 actions, and episodes terminate after the first action is taken. The reward following each action is normally distributed with a randomly generated mean and scale parameter. The reward distribution parameters are sampled from a uniform distribution on $[0, 1]$ at the start of each experimental trial.

Our second domain, GridWorld, is a discrete state and action domain that has been used in prior policy evaluation research (e.g., [[Thomas and Brunskill, 2016, Farajtabar et al., 2018]]). The domain (shown in Figure 5(b)) has $4 \times 4$ states. The agent starts from $(0, 0)$ and has the action space {left, right, up, down}. The reward is $-1$ in all non-terminal states except $(1, 1)$ where the reward is $-10$ and $(1, 3)$ where the reward is $+1$. The agent receives $+10$ in terminal state $(3, 3)$. The maximum number of steps is 100 and $\gamma = 1$ in this domain.

Our last two domains are based on the CartPole problem from OpenAI Gym [Brockman et al., 2016]. In CartPole, the agent tries to balance a pole, mounted on a cart, by moving the cart left and right.

States are given as vectors that describe the cart position and velocity and pole angle and angular velocity. Our third domain is the standard variant in which the agent controls the cart by choosing either a constant leftward force of $-1$, no force, or a constant rightward force of $+1$. Our fourth domain is a continuous control variant in which the agent can control the exact force ranging from $-1$ to $1$. In either case, the agent receives a reward of $+1$ for each time-step until termination when the pole falls or the cart moves out of bounds. The maximum number of steps is 200 and $\gamma = 0.99$ in this domain.

## C.2 Creation of Pre-trained Evaluation Policy

Each domain requires creation of an evaluation policy to serve as $\pi_e$. In the three domains with a discrete action space we use softmax policies of the form:

$$\pi_{\boldsymbol{\theta}}(a|s) \propto \frac{e^{w_a^\top \phi(s)}}{\sum_{b \in \mathcal{A}} e^{w_b^\top \phi(s)}} \tag{13}$$

where $\phi$ is a one-hot encoding operator for domains with discrete state space (Bandit and GridWorld) and a feed-forward neural network for continuous state spaces (CartPole). For ContinuousCartPole,

$$\pi_{\boldsymbol{\theta}}(a|s) := \mathcal{N}\left(a; \mathbf{w}_\mu^\top \phi(s), \left(\mathbf{w}_\sigma^\top \phi(s)\right)^2\right), \tag{14}$$

where $\phi$ is a function of the state given by a feed-forward neural network. The policy parameters, $\boldsymbol{\theta}$, denote all policy parameters. For Bandit and GridWorld, this is just the vectors $\mathbf{w_a}$ and for CartPole and ContinuousCartPole, $\boldsymbol{\theta}$ also includes the neural network weights and biases. When $\phi$ is a neural network, it is constructed with one batch normalization layer as the first layer, followed by two hidden layers, both of which have 64 hidden states and use ReLU as the activation function. We use PyTorch for neural network implementations [Paszke et al., 2019] and NumPy for linear algebra computations [Harris et al., 2020].

For all domains, we use REINFORCE [Williams, 1992] to train the policy model, and choose a policy snapshot during training as the evaluation policy, which has higher returns than the uniformly random policy, but is still far from convergence. To obtain $v(\pi_e)$, we use on-policy sampling to collect $10^6$ trajectories and compute the Monte Carlo estimate of $v(\pi_e)$.

## C.3 Off-policy Data for With Initial Data Experiments

To create an initial data set of slightly off-policy data for each domain, we create the behavior policy $\pi_b$ based on the evaluation policy $\pi_e$. For domains with discrete action space, the off-policy behavior policy is built as:

$$\pi_b(a|s) = (1 - \delta)\pi_e(a|s) + \delta \frac{1}{|\mathcal{A}|}$$

where $\delta \in (0, 1]$ controls the probability of randomly choosing an action from the action space, and otherwise sampling an action from the evaluation policy $\pi_e$. For ContinuousCartPole, the behavior policy is built as:

$$\pi_b(a|s) = \mathcal{N}\left(a; \mu_e(s), ((1 + \delta)\,\sigma_e(s))^2\right)$$

where $\mu_e(s)$ and $\sigma_e(s)$ are the output mean and standard deviation of the evaluation policy $\pi_e$ in state $s$, and $\delta$ (for $\delta > 0$) increases the standard deviation. In all domains, we use $\delta = 0.1$ and collect 100 trajectories to create the initial off-policy data set.

# D Measuring Sampling Error

One of our central claims is that ROS reduces sampling error and reducing sampling error corresponds to a reduction in MSE for policy evaluation. To evaluate this claim, we must define a metric for measuring sampling error. In this appendix, we describe two possible metrics and demonstrate how these metrics change as OS and ROS collect data for policy evaluation.

### D.1 KL-divergence of Data Collection

Our first metric (also used in the main paper), is to measure sampling error in the collected data $D$ by using KL-divergence of the empirical policy $\pi_D$ and the evaluation policy $\pi_e$. The KL-divergence between $\pi_D$ and $\pi_e$ in a particular state $s$ is defined as:

$$D_{\mathrm{KL}}(\pi_D, \pi_e) := \mathbf{E}\left[\log \frac{\pi_D(A|s)}{\pi_e(A|s)} \,\middle|\, A \sim \pi_D\right].$$

To obtain an explicit representation of $\pi_D$, we use a parametric estimate by maximizing the log-likelihood function over a parametric policy class. Specifically, we use:

$$\hat{\boldsymbol{\theta}} = \arg\max_{\boldsymbol{\theta}'} \sum_{(s,a)\in\mathcal{D}_1} \log \pi_{\boldsymbol{\theta}'}(a|s)$$

for a policy class parameterized by $\boldsymbol{\theta}$. We then use $\pi_{\hat{\boldsymbol{\theta}}}$ in place of $\pi_D$ when computing the KL-divergence. Our final sampling error metric for a data set $\mathcal{D}_1$ is:

$$D_{\mathrm{KL}}(\pi_e, D) := \sum_{(s,a)\in\mathcal{D}_1} \log \pi_{\hat{\boldsymbol{\theta}}'}(a|s) - \log \pi_e(a|s).$$

The sampling error curves of data collection **without** and **with initial data** are shown in Figure 6 and 7, respectively. We observe from these experiments that ROS can collect data with lower sampling error than OS and BPG.

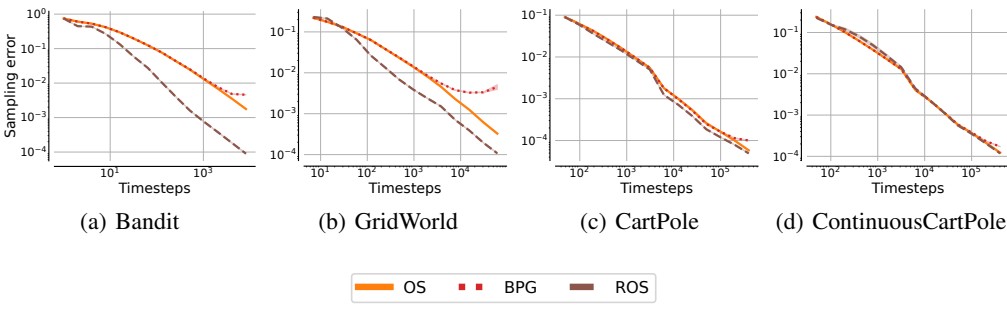

(a) Bandit     (b) GridWorld     (c) CartPole     (d) ContinuousCartPole

Figure 6: Sampling error (KL) of data collection **without initial data**. Each strategy is followed to collect data with $2^{13}\overline{T}$ steps, and all results are averaged over 200 trials with shading indicating one standard error intervals. Axes in these figures are log-scaled.

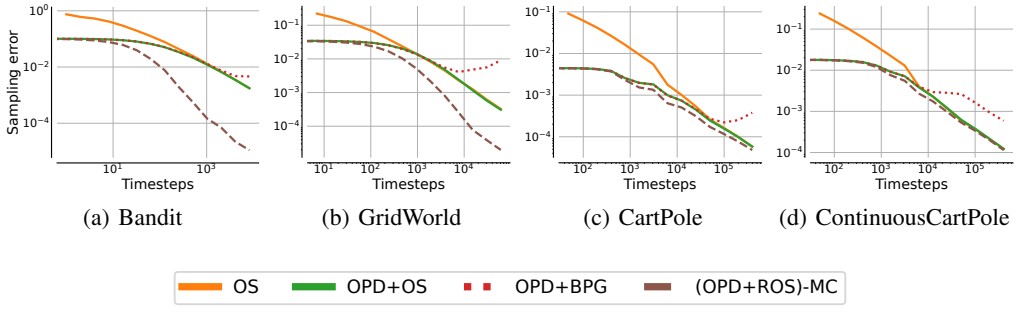

(a) Bandit     (b) GridWorld     (c) CartPole     (d) ContinuousCartPole

Figure 7: Sampling error (KL) of data collection **with initial data**. Steps, axes, trials and intervals are the same as Figure 6.

### D.2 l1-Norm of the Log-likelihood Gradient

In this section, we propose an alternative sampling error measure based on the norm of the log-likelihood gradient evaluated at $\pi_e$. If $\pi_e$ is the maximum likelihood policy under $\mathcal{D}$ then the norm of

the gradient evaluated at $\boldsymbol{\theta}_e$ will be zero when sampling error is zero. A non-zero norm indicates the parameters must change from $\boldsymbol{\theta}_e$ to maximize the log-likelihood under the observed data. We show empirically that this measure of sampling error roughly corresponds to using the KL-divergence by computing the gradient norm of log-likelihood with respect to the data collection without and with initial data, shown in Figure 8 and 9, respectively. These curves are generally consistent with their corresponding sampling error curves in Appendix D.1.

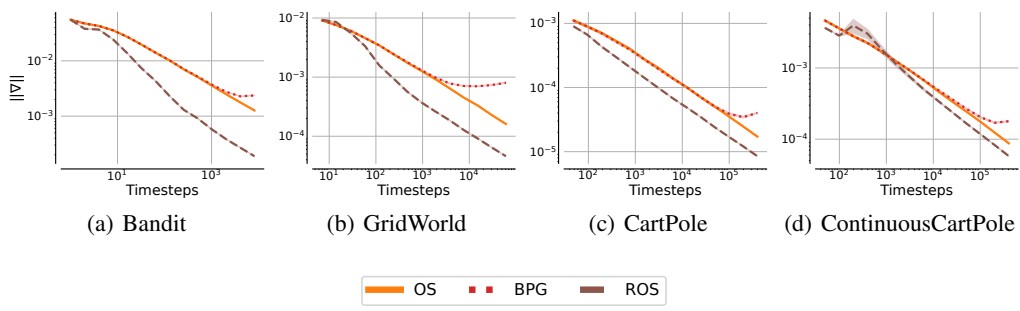

(a) Bandit      (b) GridWorld      (c) CartPole      (d) ContinuousCartPole

Figure 8: Log-likelihood gradient of the data collection **without initial data**. Steps, axes, trials and intervals are the same as Figure 6.

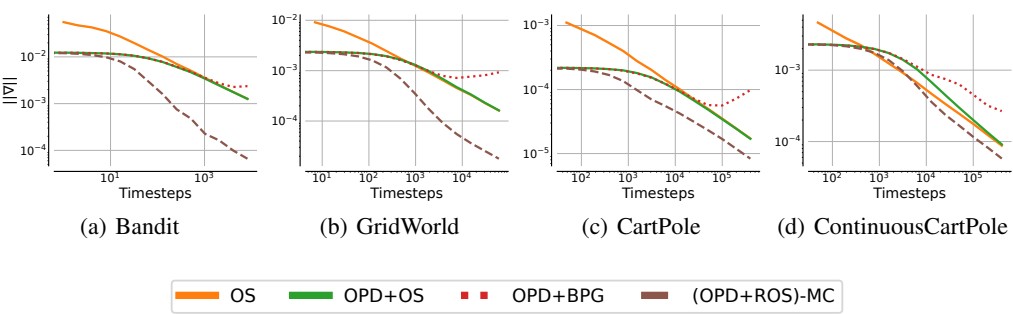

(a) Bandit      (b) GridWorld      (c) CartPole      (d) ContinuousCartPole

Figure 9: Log-likelihood gradient of the data collection with initial data. Steps, axes, trials and intervals are the same as Figure 6.

## E    Hyper-parameter Configurations

This appendix gives the hyper-parameter settings for the policy evaluation experiments. Settings for the **without initial data** experiments are given in Table 1 and settings for the **with initial data** experiments are given in Table 2. For BPG, $k$ denotes the batch-size and $\alpha$ the step-size.

| Domain | BPG - $k$ | BPG - $\alpha$ | ROS - $\alpha$ |
|---|---|---|---|
| Bandit | 10 | 0.01 | 10000.0 |
| GridWorld | 10 | 0.01 | 1000.0 |
| CartPole | 10 | 5e-05 | 10.0 |
| CartPoleContinuous | 10 | 1e-06 | 0.1 |

Table 1: Hyper-parameters for experiments **without initial data**.

| Domain | BPG - $k$ | BPG - $\alpha$ | ROS - $\alpha$ |
|---|---|---|---|
| Bandit | 10 | 0.01 | 10000.0 |
| GridWorld | 10 | 0.01 | 10000.0 |
| CartPole | 10 | 5e-05 | 10.0 |
| CartPoleContinuous | 10 | 1e-06 | 0.1 |

Table 2: Hyper-parameters for experiments **with initial data**.

# F  Numerical Results of Policy Evaluation with Initial Data

This appendix provides the numerical final values for the MSE of policy evaluation with each data collection method. Table 3 gives these values for the **without initial data** experiments and Table 4 gives these values for the **with initial data** experiments. These tables provide the numerical value corresponding to the final MSE value for each method-domain pair shown in Figures 2 and 3.

| Policy Evaluation | Bandit | GridWorld | CartPole | CartPoleContinuous |
|---|---|---|---|---|
| OS - MC | 2.06e-04 $\pm$ 2.12e-05 | 2.68e-04 $\pm$ 2.42e-05 | 2.61e-05 $\pm$ 2.40e-06 | 4.44e-05 $\pm$ 4.01e-06 |
| BPG - OIS | 9.52e-05 $\pm$ 8.83e-06 | 2.81e-04 $\pm$ 2.63e-05 | 1.20e-05 $\pm$ 1.11e-06 | 3.62e-05 $\pm$ 3.54e-06 |
| ROS - MC | 6.17e-05 $\pm$ 5.77e-06 | 1.36e-05 $\pm$ 1.33e-06 | 1.01e-05 $\pm$ 9.79e-07 | 3.29e-05 $\pm$ 3.01e-06 |

Table 3: Final MSE of policy evaluation **without initial data**. These results give the MSE for policy evaluation at the end of data collection, averaged over 200 trials $\pm$ one standard error.

| Policy Evaluation | Bandit | GridWorld | CartPole | CartPoleContinuous |
|---|---|---|---|---|
| OS - MC | 1.62e-04 $\pm$ 1.45e-05 | 2.68e-04 $\pm$ 2.42e-05 | 2.61e-05 $\pm$ 2.40e-06 | 4.44e-05 $\pm$ 4.01e-06 |
| (OPD + OS) - MC | 1.48e-04 $\pm$ 1.41e-05 | 2.80e-04 $\pm$ 2.82e-05 | 2.69e-05 $\pm$ 2.70e-06 | 4.27e-05 $\pm$ 3.72e-06 |
| (OPD + OS) - (WIS + MC) | 1.56e-04 $\pm$ 1.70e-05 | 2.65e-04 $\pm$ 2.40e-05 | 2.59e-05 $\pm$ 2.51e-06 | 6.02e-05 $\pm$ 8.51e-06 |
| (OPD + BPG) - OIS | 6.86e-05 $\pm$ 7.46e-06 | 9.20e-04 $\pm$ 6.27e-04 | 1.52e-05 $\pm$ 1.53e-06 | 1.12e-03 $\pm$ 9.58e-04 |
| (OPD + ROS) - MC | **4.78e-05 $\pm$ 4.81e-06** | 2.35e-06 $\pm$ 2.42e-07 | 9.60e-06 $\pm$ 9.35e-07 | 3.27e-05 $\pm$ 3.21e-06 |

Table 4: Final MSE of policy evaluation **with initial data**. These results give the MSE for policy evaluation at the end of data collection, averaged over 200 trials $\pm$ one standard error.

# G  Median and Inter-quartile Range of Policy Evaluation

In this appendix, we present the computed median and interquartile range for the squared error of policy evaluation both **with** and **without initial data** across all four domains. In the main paper we present the mean squared error and standard error as is typical in the policy evaluation literature. Here, we include the median and interquartile ranges as they are more robust statistics. We give these results for completeness; qualitatively, they leave the conclusions from the main paper unchanged.

Figure 10 shows the median and interquartile ranges for policy evaluation **without initial data**. Figure 11 gives the same for policy evaluation **with initial data**. From these figures, we can observe that all data collection methords have a similar inter-quartile range and ROS lowers the median of the squared error compared to OS.

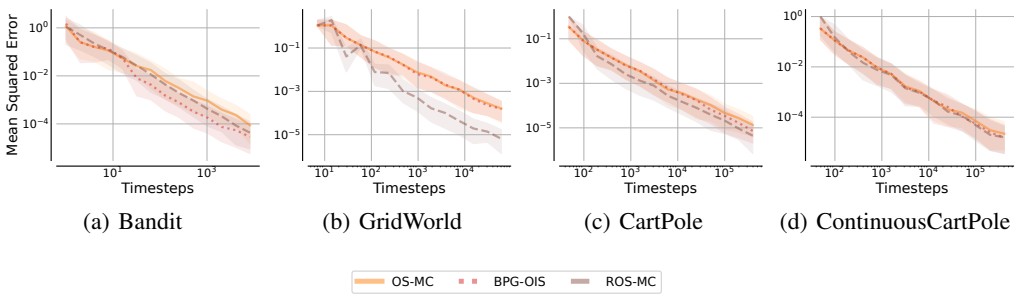

(a) Bandit    (b) GridWorld    (c) CartPole    (d) ContinuousCartPole

OS-MC    BPG-OIS    ROS-MC

Figure 10: Median and inter-quartile range of squared error (SE) of policy evaluation **without initial data**. The lines in these figures denote the median of squared error over 200 trials, and the shading indicates the interquartile range. Axes in these figures are log-scaled.

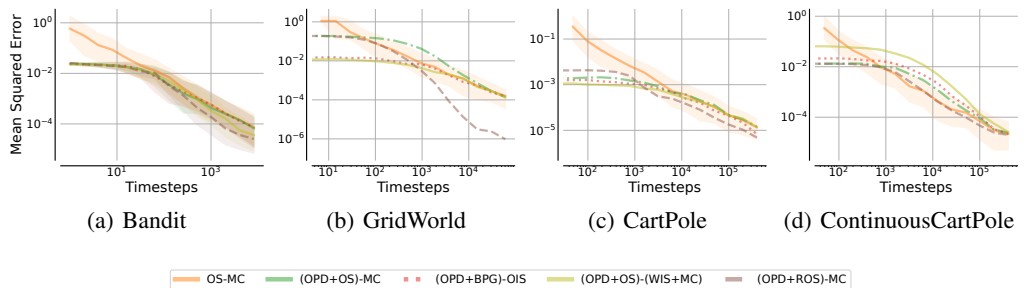



OS-MC    (OPD+OS)-MC    (OPD+BPG)-OIS    (OPD+OS)-(WIS+MC)    (OPD+ROS)-MC



Figure 11: Median and inter-quartile range of squared error (SE) of policy evaluation **with initial data**. Axes, trials and intervals are the same as Figure 10.

## H Environment and Policy Sensitivity in GridWorld

In the main paper, we evaluated the relative MSE of ROS compared to OS in the Bandit environment under different settings of the reward scale and variance and the stochasticity of $\pi_e$. In this appendix, we repeat the same study in the GridWorld environment. The original GridWorld domain has fixed rewards for each state. We vary these by either 1) multiplying by a fixed factor (referred to as the mean factor) or 2) replacing the deterministic reward with a reward sampled uniformly from $[-f_{\text{scale}}, f_{\text{scale}}]$ where $f_{\text{scale}}$ determines the reward variance. We then follow OS or ROS ($\alpha = 1000$) to collect 1000 trajectories, and perform Monte Carlo estimation. The relative MSEs between OS and ROS are shown in Figure 12(a). Results show an identical trend to the Bandit environment: a larger reward scale increases the amount of improvement because small amounts of sampling error can lead to larger amounts of error; larger reward variance decreases the amount of improvement because the MSE becomes dominated by variance in the reward.

We also create different evaluation policies using $\epsilon$-greedy policies with $\epsilon$ ranging from 0 to 1. We then perform the same policy evaluation as above and compute the relative MSEs, shown in Figure 12(b). As with the Bandit domain, we see that greater entropy in $\pi_e$ generally leads to a wider margin of improvement between ROS and OS.

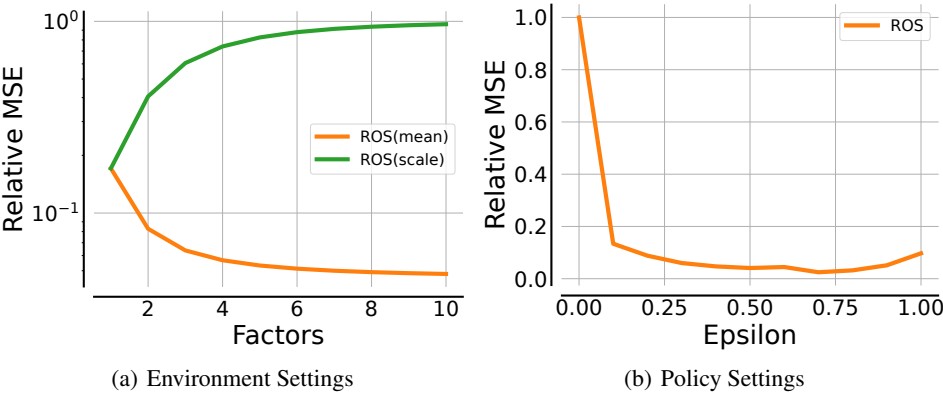

(a) Environment Settings           (b) Policy Settings

Figure 12: Improvement of ROS compared to OS with different settings in GridWorld. Axes and trials are the same as Figure 4.

## I Mean Squared Error of Policy Evaluation with WRIS and FQE

This appendix presents preliminary results on combining ROS with estimators specifically designed for the off-policy setting instead of the Monte Carlo estimator. Specifically, we used weighted regression importance sampling (WRIS) [Hanna et al., 2021] and fitted q-evaluation (FQE) [Le et al., 2019]. We study these estimators in the **without initial data** setting and show their MSE with varying

amounts of data. Figure 13 shows the MSE of different data collection methods for WRIS across the different domains. Figure 14 shows the same except with FQE as the estimator.

We observe in tabular domains that OS cannot obtain the same level MSE as ROS, even with the help of WRIS, which is designed to correct sampling error during the value estimation stage. This shows the importance of reducing sampling error during the data collection stage. The same pattern can also be observed when using FQE. In non-tabular domains, the performances of off-policy evaluation methods with different data collection methods are very similar. However, WRIS usually requires larger sizes of data to make accurate estimation, and can only achieve similar MSE as ROS when collecting a large amount of data. When using FQE in non-tabular domains, data from ROS can generally enable lower MSE than OS, although this improvement is very small.

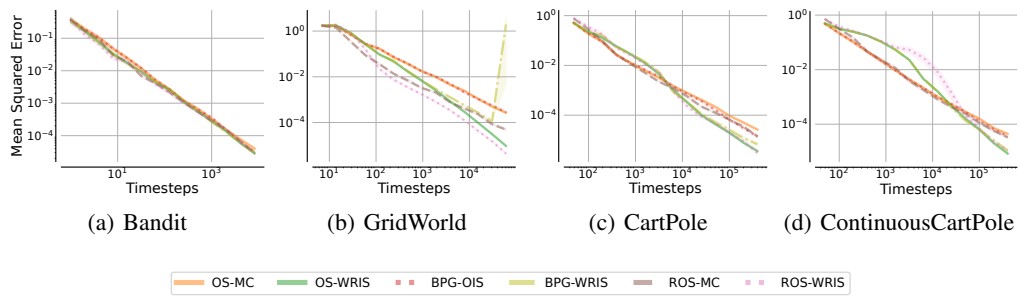

Figure 13: MSE of WRIS **without initial data**. Steps, axes, trials and intervals are the same as Figure 2.

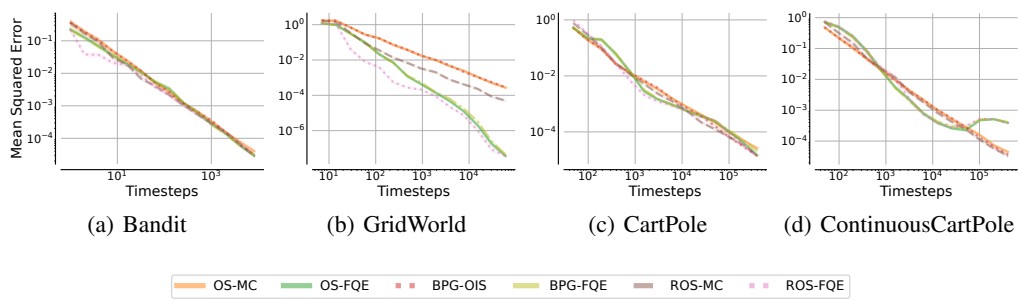

Figure 14: MSE of FQE **without initial data**. Steps, axes, trials and intervals are the same as Figure 2.