# OpenReview forum: "Robust On-Policy Sampling for Data-Efficient Policy Evaluation in Reinforcement Learning"
_NeurIPS.cc/2022/Conference — NeurIPS 2022 Accept_

### Official Review · Reviewer_gCc9 · 2022-07-07

**Rating:** 4
**Confidence:** 3
**Soundness:** 2 fair
**Presentation:** 3 good
**Contribution:** 2 fair

**Summary:**

The authors propose a method of more efficient policy evaluation, that requires fewer samples from the environment relative to Monte Carlo sampling (running directly in environment), as well as an alternative baseline method BPG. The central idea of the paper is that a marginal log-likelihood loss can be treated as a gradient in the direction of executing actions that appear more often in evaluation trajectories seen so far. Therefore, the negative of the gradient can direct the policy in the direction of executing actions that have been seen less often, which causes our empirical action frequency to stay closer to the true distribution compared to sampling actions naively.

This approach is compared both assuming no existing episodes of interaction, and with episodes of interaction - if there is initial data then it is used to compute the gradient for future interactions in the environment / provide an initial set of episodes for averaging episode return.

**Questions:**

What do the authors see as potential paths for applying the problem to larger dimensional MDPs? It seems the core of this paper is whether we expect this to work in larger MDPs, because it is only in those MDPs where you need function approximation to work. But it is also in larger MDPs where we'd expect it to be difficult to adjust policy behavior by applying a small number of gradients to the policy.

**Limitations:**

Yes

**Strengths And Weaknesses:**

The insight that the log-likelihood can be treated as a gradient in the direction of selecting rarer actions is interesting. I do have some concerns about the significance of the method. The method inherently relies on having the ability to sample enough on-policy trajectories to apply meaningful corrections to the current empirical distribution of episode trajectories. I would assume this requires a fairly substantial number of episodes as you try to apply the method to higher dimensional settings. It also seems like this approach becomes more data hungry as you increase the size of the existing initial data, because it will take more on-policy samples to apply the relevant corrections.

There is a theoretical justification of the approach, but the argument for it has some fairly strong cavaets - it is both based on a tabular policy, and requires using a learning rate of $\alpha \to \infty$, such that ROS will always pick an action that is undersampled. This does give a sequence of random variables that strays from the true distribution less, but it's a pretty extreme argument and it's hard to see how this proof says anything about the continuous, non-tabular case with much smaller learning rates. It feels a good deal more extreme than other tabular RL proofs. Most of the time you can tell the proof will not work directly in a continuous setting, but you can see a vague throughline of how it could work assuming a nice enough function approximator. I don't see valid throughline if you need to start from assuming an infinite learning rate.

Additionally, I feel there are errors in the proof used. In the proof of Lemma 1 in Appendix B, the paper says

"The ROS update is given by $\theta'_a = \alpha(\frac{\theta}{\alpha} - term)$. Taking the limit as $\alpha \to \infty$ we see that $\frac{\theta}{\alpha}$ is negligible and thus the parameter is $\theta'_a = \alpha(term)$"

I don't see why this step is justified, it relies on $\lim_{\alpha\to\infty} \alpha \cdot \frac{\theta}{\alpha} = 0$, which is definitely not correct, we cannot cancel this term to 0 if it's getting multiplied by a term going to infinity.

In the proof of Theorem 2, the authors could also be clearer on why $Q_m$ can be treated as O(1) (because the n-th derivative of the KL-divergence at a fixed point $\pi_e$ should still be constant with respect to number of data points $m$). In general it would have also been nicer if size of action space $|A|$ was in the bounds in some way, because this does matter to the convergence rate if we wished to apply this method to different MDPs.

(I suspect that although the proofs have holes, the general results are still correct.)

The method does outperform BPG asymptotically - my concern is that by the time the asymptotics show up, you are already collecting enough on-policy episodes to make the difference not very applicable. Almost all results are in discrete domains with learning rates $\alpha = 10$ to $10^4$ and the results in the one continuous problem with reasonable learning rate (CartPoleContinuous) did not look as exciting.

---

> ### Author Response · Authors · 2022-08-02
> **Response to Reviewer gCc9**
>
> Thank you for reviewing the paper and particularly for carefully checking the proofs and pointing out some issues which we address below and in the revision.
>
> > The method inherently relies on having the ability to sample enough on-policy trajectories to apply meaningful corrections to the current empirical distribution of episode trajectories. I would assume this requires a fairly substantial number of episodes as you try to apply the method to higher dimensional settings. It also seems like this approach becomes more data hungry as you increase the size of the existing initial data, because it will take more on-policy samples to apply the relevant corrections.
>
> Note that Figure 2 uses time-steps on the x-axis and not trajectories. We see that ROS is able to meaningfully impact sampling error and MSE even with just a handful of trajectories.
>
> You are correct about a larger initial data set though what you say is true for OS as well as ROS. If the initial data set is large and has a large distribution shift from the desired on-policy distribution then ROS will need to collect more data than if the initial data set was small or had a smaller distribution shift. The same is true for OS. Our main point is that ROS will require less than OS and Figure 3 confirms this point.
>
> > I don't see valid throughline if you need to start from assuming an infinite learning rate.
>
> ROS with any non-zero, finite learning rate can be viewed as interpolating between on-policy sampling and always taking the action that is most under-sampled. Theorem 2 characterizes both extremes. In theory, using a smaller learning rate corresponds to keeping probability on sub-optimal actions, though in practice we may wish to do so when the tabular assumption is violated.
>
> > Errors in proofs
>
> Thanks for pointing out the error regarding the parameter update. We have corrected the statement in the revision and the claim (that ROS always takes the most under-sampled action) remains correct.
>
> We also have clarified the statement on higher-order derivatives of the empirical KL-divergence based on your suggestion. Unfortunately, we lack an immediate generalization to include the size of the action space in these results – we agree this would be a useful result to consider in a future investigation.
>
> > the results in the one continuous problem with reasonable learning rate (CartPoleContinuous) did not look as exciting.
>
> It’s true that in the “without initial data” setting, we only see a small improvement using ROS on CartPoleContinuous. However, please note that in the “with initial data” setting, the gap widens between ROS and OS (even when OS is able to replace the initial off-policy trajectories with on-policy trajectories). We find this result exciting as it shows that ROS can take an initially biased dataset and collect the right trajectories to make it look as-if the evaluation policy had collected all trajectories in the first place.
>
> > What do the authors see as potential paths for applying the problem to larger dimensional MDPs? It seems the core of this paper is whether we expect this to work in larger MDPs, because it is only in those MDPs where you need function approximation to work. But it is also in larger MDPs where we'd expect it to be difficult to adjust policy behavior by applying a small number of gradients to the policy.
>
> As noted in the reply to all reviewers, we see the core of the paper as exploring how non-iid off-policy can produce data that is more on-policy than the data produced by on-policy sampling. ROS is a first method in this space and the core message is not diminished if ROS as-introduced does not immediately work well on larger dimensional MDPs.
>
> That said, ROS has a really nice property – that it can produce empirical distributions that match a given distribution without explicitly tracking the empirical distribution – and this property is appealing for larger MDPs. We suspect that ROS with a well-tuned step-size will be able to perform well in larger MDPs but that adaptive step-size methods are a promising direction for robust performance. A natural gradient [2] style update as in TRPO [3] could also be useful to ensure that a single policy update makes a meaningful change in action distribution space and not just policy parameter space.
>
> [2] A Natural Policy Gradient. Kakade. 2001
> [3] Trust Region Policy Optimization. Schulman et al. 2015.

---

> > ### Comment · Reviewer_gCc9 · 2022-08-08
> > **Response**
> >
> > Thanks for the reply.
> >
> > > Figure 2 uses time-steps on the x-axis
> >
> > Thanks for clarifying this. It would be helpful if the steps/episode for each domain were mentioned here, rather than in the supplemental material.
> >
> > > ROS with any non-zero, finite learning rate can be viewed as interpolating between on-policy sampling and always taking the action that is most under-sampled. Theorem 2 characterizes both extremes. In theory, using a smaller learning rate corresponds to keeping probability on sub-optimal actions, though in practice we may wish to do so when the tabular assumption is violated.
> >
> > Could you clarify what you mean by Theorem 2 characterizing both extremes? I thought Theorem 2 was talking about assuming Assumption 1, the $\alpha \to \infty$ assumption. I agree that smaller learning rates will cause you to update towards less sampled actions less strongly.
> >
> > > However, please note that in the “with initial data” setting, the gap widens between ROS and OS (even when OS is able to replace the initial off-policy trajectories with on-policy trajectories).
> >
> > These results do seem more encouraging. Why does MSE start so much higher for OS-MC compared to OPD+OS-MC, for example? By my reading, the only difference is that (OPD+OS)-MC = 100 episodes off-policy + on-policy samples, and OS-MC = 100 on-policy episodes + more on-policy samples - it seems strange that (OPD+OS)-MC starts with a lower MSE in the cartpole envs?

---

> > > ### Author Response · Authors · 2022-08-09
> > > **Re: Response**
> > >
> > > Thank you for taking the time to read our response and seek clarification.
> > >
> > > > It would be helpful if the steps/episode for each domain were mentioned here, rather than in the supplemental material.
> > >
> > > We agree and will update the paper with these details.
> > >
> > > > Could you clarify what you mean by Theorem 2 characterizing both extremes?
> > >
> > > Apologies for wording this ambiguously. By “characterizes” we mean that Theorem 2 gives the convergence rate when we always take the most under-sampled action ($\alpha=\infty$) and the convergence rate when we simply sample from $\pi_e$ ($\alpha=0$).
> > >
> > > > Why does MSE start so much higher for OS-MC compared to OPD+OS-MC, for example?
> > >
> > > We see how our description in the paper is a little confusing and will clarify the paper. *The off-policy data is not included in the horizontal axis total* and so OPD+OS-MC and OPD+ROS-MC have access to 100 extra trajectories that initially lower MSE at the expense of injecting a large bias into the estimate. OPD+OS-MC struggles to reduce this bias while OPD+ROS-MC is able to through data collection.
> > >
> > > *The 100 on-policy trajectories are included for OS-MC*, however, our point still stands if we adjust the OS-MC curve to remove them. Note that the log scale horizontal axis means this adjustment is not a constant leftward shift of the OS-MC curve but a leftward shift that becomes smaller as you move right in the figure. Initially, this adjustment will lead to OS-MC having lower MSE than OPD+ROS-MC as OPD+ROS-MC overcomes the initial bias. However, around 10^4 steps (approximately 200 trajectories) the order of the methods switches and OS-MC will have higher MSE even with the off-policy trajectories replaced with on-policy trajectories. It's at this point that ROS has taken an initially biased dataset and collected the right trajectories to make it look as-if the evaluation policy had collected all trajectories in the first place.
> > >
> > > We will clarify the figures in the main paper text and highlight the key points that the OPD+*-MC methods are initially biased and OPD+ROS-MC is able to first correct this bias and then lower MSE compared to OS-MC which does not have the initial bias to overcome.

---

### Official Review · Reviewer_cwVN · 2022-07-11

**Rating:** 6
**Confidence:** 3
**Soundness:** 3 good
**Presentation:** 4 excellent
**Contribution:** 3 good

**Summary:**

The authors identify a deficiency in on-policy data collection for policy evaluation; when data are sampled i.i.d., new samples do not take into consideration the possible bias of the previous samples. The authors propose adaptive, *off-policy* sampling as a means to actively correct for this bias in a feedback-based manner. Because computing the exact policy for the bias correction is not feasible when using function approximation, the authors propose Robust On-policy Sampling (ROS), which iteratively perturbs a stochastic policy via gradient ascent to iteratively approximate the required sampling policy. The authors prove some convergence results for their method and demonstrate that it can reduce the expected mean squared error (MSE) for policy evaluation in several low-dimensional control tasks.

**Questions:**

1. Is there a possible relationship between the theoretical KL divergence bound in Theorem 2 and the expected, empirical MSE? I understand that, intuitively, a lower KL divergence would generally lead to a lower MSE in practice, but as it stands now, Theorem 2 seems to be relatively disconnected from the empirical results of the paper. I would be willing to raise my score if the authors can strengthen the connection between theory and practice here (e.g., possibly by relaxing/removing Assumption 1).
1. More of a suggestion than a question: Do you think an adaptive gradient method (e.g., RMSProp, Adam) might help mitigate some of the hyperparameter sensitivity observed in Section 6.3?


**Strengths And Weaknesses:**

**Strengths**
- The authors consider an interesting and relevant problem for on-policy evaluation: Given a batch of previously collected (possibly biased) on-policy data, what *off-policy* sampling strategy should be employed to produce better estimates of the *on-policy* expected returns under the target policy $\pi$? As the paper correctly points out, simply sampling more i.i.d. data using the target policy is a suboptimal strategy.
- The proposed algorithm ROS is an intuitive solution to the above problem. Given that some actions may be underrepresented in the past, it makes sense that their probabilities should be slightly increased in the future. Using gradient ascent on a stochastic policy is an elegant way to do this without requiring the explicit calculation of an analytical solution, which is crucial for non-tabular environments.
- The empirical performance of the proposed method appears to be favorable when compared to a variety of baselines, although I am not familiar enough with the related work to determine the relative strengths of the baselines.
- Overall, the paper is very well written.

**Weaknesses**
- Some of the theoretical results are somewhat weak:
  - Proposition 1, which states that pooling two i.i.d. datasets together is biased unless the first dataset is unbiased, seems obvious and unnecessary; of course, if the first dataset's mean is wrong, then it would require an infinite amount of additional sampling to counteract its bias.
  - Assumption 1 (the MDP is a directed acyclic graph) is a very strong assumption that I think would rarely be satisfied in practice.
  - Theorem 1 establishes the consistency of the ROS estimator. However, the i.i.d. on-policy estimator (which the paper claims is inferior) is also a consistent estimator! I find this confusing, because the main thesis of the paper is that a consistent estimator (i.e., i.i.d. on-policy sampling) is, alone, not sufficient for good performance with finite trajectories. Having consistency/convergence as a theorem therefore seems to overstate its importance; perhaps it would be better as a lemma to Theorem 2?
- I was a bit disappointed to see that the main convergence result (Theorem 2) provides a stochastic bound on the KL divergence between the maximum-likelihood policy $\pi_D$ and the true policy $\pi$, rather than the policy-evaluation MSE. As far as I understood, the authors’ claim in the abstract/introduction is that ROS should lead to lower MSE. Although this is shown empirically, which is great, the paper would be much stronger if the theory also supported this claim. In my opinion, it is not as interesting that ROS is able to estimate the policy $\pi$ with high accuracy, but more so that, by doing this, it can achieve more-accurate estimates of $V^\pi$ or $Q^\pi$.
- The tested environments are relatively small, comprising a bandit, gridworld, and two cartpole variations. I think adding at least one high-dimensional environment – even if it were just one Atari game or the like – would make the experiments much more convincing. At present, it is hard to gauge how scalable the results are, even though the ROS algorithm itself looks scalable.

**Minor Edits**
- Line 122: “is ran” $\rightarrow$ “is run.”
- Line 176: “can be ran” $\rightarrow$ “can be run.”
- Line 191: Should it be $\theta \in \mathbb{R}^d$, not $\theta \subseteq \mathbb{R}^d$?

---

> ### Author Response · Authors · 2022-08-02
> **Response to Reviewer cwVN**
>
> Thank you for taking the time to review the paper and your appreciation of ROS and its presentation. We have made a couple of clarifications and small changes to the paper that we hope will address the reviewer’s concerns regarding how the theoretical results connect to practice.
>
> > Proposition 1, which states that pooling two i.i.d. datasets together is biased unless the first dataset is unbiased, seems obvious and unnecessary; of course, if the first dataset's mean is wrong, then it would require an infinite amount of additional sampling to counteract its bias.
>
> We agree that Proposition 1 may seem obvious, however, we respectfully submit that including it is important for illustrating how known, available data changes how we should discuss the statistical properties of an estimator. It is not one of the main theoretical contributions of our paper.
>
> > Assumption 1 (the MDP is a directed acyclic graph) is a very strong assumption that I think would rarely be satisfied in practice.
>
> See our reply to your question below.
>
> > Theorem 1 establishes the consistency of the ROS estimator. However, the i.i.d. on-policy estimator (which the paper claims is inferior) is also a consistent estimator! I find this confusing, because the main thesis of the paper is that a consistent estimator (i.e., i.i.d. on-policy sampling) is, alone, not sufficient for good performance with finite trajectories. Having consistency/convergence as a theorem therefore seems to overstate its importance...
>
> Theorem 1 is important for establishing the soundness of ROS. We view consistency as a minimal property that any estimator should satisfy and we completely agree that on-policy sampling is not inferior in this respect. Both OS and ROS satisfy this minimal property. However, note that the consistency of OS does not imply anything about its data efficiency. While OS is consistent, our main thesis is that it is inefficient.
>
> > Main result bounds KL divergence and not MSE
>
> See our response to your question below. We see your point and have updated the paper with a new result that relates the KL divergence to policy evaluation error.
>
> > Environment complexity
>
> See comment to all reviewers.
>
> > Is there a possible relationship between the theoretical KL divergence bound in Theorem 2 and the expected, empirical MSE? I understand that, intuitively, a lower KL divergence would generally lead to a lower MSE in practice, but as it stands now, Theorem 2 seems to be relatively disconnected from the empirical results of the paper. I would be willing to raise my score if the authors can strengthen the connection between theory and practice here (e.g., possibly by relaxing/removing Assumption 1).
>
> We have updated the paper with a new theoretical result that strengthens the connection between the empirical policy convergence rate and the rate at which policy evaluation error decreases. As a result sketch, we can show that the squared error is upper-bounded by 1) the KL-divergence between the empirical policy and the evaluation policy and 2) the KL-divergence between the empirical state distribution and expected state distribution under the evaluation policy. Term (1) decreases at a faster rate using ROS compared to OS (by Theorem 2). Term (2)’s convergence depends on the MDP’s transitions and initial state distribution and thus we cannot place a definite order on ROS and OS for this term. If the environment is highly stochastic then the empirical transitions and initial state distribution prevent a faster rate than 1/m as states are always sampled conditionally iid from these distributions. These observations are corroborated by Figure 4c which shows that as environment stochasticity increases, ROS improves upon OS less in terms of relative MSE.
>
> We can also remove Assumption 1 as requested as it is actually unnecessary due to the finite-horizon MDP setting we consider. In a finite-horizon MDP, it is common practice to include the current time-step in the state definition (see the end of Section 1.2 in [1] for more discussion); doing so is also theoretically necessary for the transition probabilities to be Markov. Thus, Theorem 1 goes through without Assumption 1 and we have removed Assumption 1 in the submitted revision.
>
> > More of a suggestion than a question: Do you think an adaptive gradient method (e.g., RMSProp, Adam) might help mitigate some of the hyperparameter sensitivity observed in Section 6.3?
>
> This is an interesting suggestion. We have begun to look at adaptive step-size methods but not more complex adaptive methods as you suggest. We see some improvement when increasing the step-size over time because, intuitively, once a lot of data has already been collected it takes a larger policy change to significantly affect the empirical distribution. For simplicity we chose to focus on constant step-sizes in this paper but agree this direction should receive more attention.
>
> [1] Reinforcement Learning: Theory and Algorithms. Agarwal et al. 2022.

---

> > ### Comment · Reviewer_cwVN · 2022-08-07
> > **Re: Response to Reviewer cwVN**
> >
> > I thank the authors for the updated theoretical contributions. Theorem 3 makes the paper much stronger by establishing an explicit relationship between KL divergence and MSE, which I think is important to motivate the practical usage of the method.
> >
> > I still feel that adding a more-complex environment would help eliminate any doubts about the scalability of ROS, as was also expressed by some of the other reviewers, but I do agree with the authors that the given empirical results demonstrate the effectiveness of the method with function approximation.
> >
> > My final concern is still regarding Assumption 1. Can you please elaborate on what you mean by
> > > In a finite-horizon MDP, it is common practice to include the current time-step in the state definition; doing so is also theoretically necessary for the transition probabilities to be Markov.
> >
> > I understand that including the timestep is needed for the agent to know when the episode will terminate (and therefore to estimate returns), but in either case, the underlying transition probabilities are always Markov. It seems to me that you are now just using the finite-horizon assumption to impose a DAG-like structure on the MDP anyway, so you can delete Assumption 1 without changing the theory. Is this not the case?

---

> > > ### Author Response · Authors · 2022-08-08
> > > **Re: Re: Response to Reviewer cwVN**
> > >
> > > Thank you for taking the time to read our response and seek clarification. We’re glad you found Theorem 3 to make the paper much stronger; thanks again for suggesting this point.
> > >
> > > Regarding your question, yes, exactly. We aren’t getting rid of the need for DAG structure but rather arguing that it is already imposed under the widely used MDP setting we assumed and so the need for a special assumption is unnecessary. Our comment on Markov transitions was regarding the terminal state probability which is time-dependent in a finite-horizon MDP.  To clarify the setting in the text, our revision added “Since we assume a finite-horizon, we assume the state definition implicitly includes temporal information.” We then reference this statement in the revised proof of Theorem 1.
> > >
> > > In case you are concerned that Theorem 1 may only hold in special circumstances, regardless of MDP structure, the empirical transition probabilities converge to the true transition probabilities by the law of large numbers and the empirical action probabilities converge to the true probabilities under $\pi$ by Lemma 2. Thus, intuitively Theorem 1 likely holds for fully general MDPs, however, we have only included the claim for which we currently have a full proof. We also wish to highlight that we did not include time as part of the state in any experiments and still see empirically consistent estimates from ROS and decreasing sampling error.

---

> > > > ### Comment · Reviewer_cwVN · 2022-08-08
> > > > **Updated score**
> > > >
> > > > Thank you for the explanation. After thinking about this more, I do like the justification of a DAG using the finite-horizon setting. It makes the assumption feel more realistic, which was my major issue with it originally.
> > > >
> > > > My final, very minor concern is that this assumption might not be emphasized enough now. Readers might not remember the sentence about the temporal information in Section 3.1 by the time they get to the theorems. I would recommend reiterating that fact somewhere in Section 5.2 when you present Theorem 1, possibly by using a \begin{assumption} environment like you had before.
> > > >
> > > > Anyway, this is primarily a presentation matter now, and I like the new theoretical results. I have updated my paper score. Thank you for addressing my concerns.

---

> > > > > ### Author Response · Authors · 2022-08-08
> > > > > **Re: Updated score**
> > > > >
> > > > > Thank you very much for the additional comment and for raising your score. We will re-emphasize the assumption in 5.2 as you suggest.

---

### Official Review · Reviewer_C33h · 2022-07-13

**Rating:** 8
**Confidence:** 3
**Soundness:** 4 excellent
**Presentation:** 4 excellent
**Contribution:** 4 excellent

**Summary:**

The paper proposes using non-i.i.d. off-policy sampling to improve the efficiency of on-policy RL algorithms.
Specifically, it suggests using a behaviour policy that takes into account past actions to choose future actions to cause the observed data to more closely match the on-policy distribution than simply relying on the law of large numbers.
The paper then presents theoretical and empirical support for the proposed approach.

**Questions:**

1. The definition of a policy is that of a Markov policy, but the paper proposes using a behaviour policy that takes into account more information than just the current state/features.
1. Why does ROS only take into account immediate sampling error, and not sampling error in future states via TD-like bootstrapping? Is it unlikely to make a big difference for the extra complication? Could this drag the behaviour policy further away from the on-policy state distribution?
1. The figures are very small, and some of the colours are quite similar. I would suggest making the figures larger, possibly by using a shared Y axis. Also consider changing the colours, bolding the lines, and/or using arrows and labels instead of a legend to reduce the reliance on colour to tell which algorithm is which.
1. In Figure 2, how is the policy evaluated? Monte Carlo?
1. Would ROS work with a changing target policy?

### Minor suggestions:
- line 33: "upmost" should be "utmost"

**Limitations:**

Yes, the paper adequately addressed the limitations and societal impact of the proposed algorithm.

**Strengths And Weaknesses:**

### Originality
To the best of my knowledge, the proposed algorithm and contributions are new.

### Quality
The paper seems very well-done. I don't have any complaints about quality.

### Clarity
The paper is written very clearly, with intuitive examples provided to communicate important points to the reader. I was pleasantly surprised by how readable the paper is.

### Significance
The proposed strategy tackles an important problem: the poor sample efficiency of on-policy algorithms. It also seems very general, and therefore could be quite significant.

---

> ### Author Response · Authors · 2022-08-02
> **Response to Reviewer C33h**
>
> Thank you for your kind comments with respect to our contributions and presentation. In particular, we are happy the reviewer appreciated the generality of the approach.
>
> > The definition of a policy is that of a Markov policy, but the paper proposes using a behaviour policy that takes into account more information than just the current state/features.
>
> Implicitly, yes, the behavior policy takes into account all past data. Note however that after the ROS update, we ultimately select an action from a Markov policy.
>
> > Why does ROS only take into account immediate sampling error, and not sampling error in future states via TD-like bootstrapping? Is it unlikely to make a big difference for the extra complication? Could this drag the behaviour policy further away from the on-policy state distribution?
>
> This is definitely an interesting direction that we have not fully explored. Reducing immediate sampling error will lower sampling error in future states as the distribution of future states depends on the distribution of actions taken in the current state. However, it is still possible that a different policy update that explicitly considers sampling error in the future will be even more effective.
>
> > The figures are very small, and some of the colours are quite similar. I would suggest making the figures larger, possibly by using a shared Y axis. Also consider changing the colours, bolding the lines, and/or using arrows and labels instead of a legend to reduce the reliance on colour to tell which algorithm is which.
>
> Thanks for the suggestion; we have updated our main figures to address and will have all updated for the camera-ready.
>
> > In Figure 2, how is the policy evaluated? Monte Carlo?
>
> Yes, both ROS and OS use the Monte Carlo estimate (Eq. 2). BPG uses the ordinary importance sampling estimate which is a fundamental part of BPG.
>
> > Would ROS work with a changing target policy?
>
> Yes, this extension should be straightforward and we are particularly excited about this possibility as a means to integrate ROS with on-policy policy improvement algorithms as a follow-up work to this paper.

---

### Official Review · Reviewer_cYQz · 2022-07-13

**Rating:** 6
**Confidence:** 3
**Soundness:** 2 fair
**Presentation:** 2 fair
**Contribution:** 2 fair

**Summary:**

In this work, authors attempt to improve sample efficiency as well as accuracy of Monte-Carlo policy evaluation by an interplay between on-policy and off-policy data collection. The key idea being on-policy sampling may not guarantee exact policy state-action visitiation distribution in finite interactions. This could be corrected by doing off-policy sampling while tracking the visitation distribution to ensure on-policy state-action density distribution. To elaborate, they collect some data using on-policy sampling, thereafter, they create a new behavior policy with higher likelihood of under-sampled actions.

Their method of “Robust On-policy Sampling (ROS)”  covers larger MDPs by assuming parametrized and differential behavior policy representation. This policy is adjusted via gradient descent to increase probabilities for low-sampled actions.


**Questions:**

I would recommend authors to evaluate their method over  Atari or Gym-based mujoco environments. This might show significant improvement in optimal policy convergence in those cases which are generally sample inefficient.

Also, can your work be adopted to deterministic parameterized policies?

**Ethics Review Area:**

["I don’t know"]

**Strengths And Weaknesses:**

Originality: Authors attempt to improve Monte-Carlo accuracy and sample efficiency in a novel way. They do have classic baselines as well enough coverage of related work.

Quality: Yes, the proposal is technically reasonable. They also provide theoretical analysis and show experimental results in simpler environments to build their case. Authors did show sensitivity of their method to step-size(hyper-parameter)

Clarity: Yes, the paper is well written and requires minor corrections only

- PE notation is not defined in eq. (1)
- MC notation is also not defined in eq. (2)

Significance: Authors open up an interesting problem space and the suggested ideas could be borrowed by others. However, the experiments done by authors are in simpler environments, where significant improvements are only found in Gridworld environment with discrete space. This dilutes the practical importance of the method.

---

> ### Author Response · Authors · 2022-08-02
> **Response to Reviewer cYQz**
>
> Thank you for your kind comments and appreciating the new problem space this paper aimed to open up. We’ve answered your questions below. Your minor comments are addressed in our revision; thanks for pointing these out!
>
> > I would recommend authors to evaluate their method over Atari or Gym-based mujoco environments. This might show significant improvement in optimal policy convergence in those cases which are generally sample inefficient.
>
> See reply to all reviewers.
>
> > Also, can your work be adopted to deterministic parameterized policies?
>
> This is an interesting question we had not considered as many on-policy RL algorithms only use stochastic policies and so ROS is designed with stochastic policies in mind. However, the general objective of adapting the behavior policy to minimize sampling error could be potentially adapted to deterministic policies. In this case we would be attempting to directly minimize sampling error in the distribution of states visited by the policy.

---

### Author Response · Authors · 2022-08-02
**Response to all reviewers**

Thank you to all the reviewers for their effort in reviewing our paper, kind comments, and suggestions for improvement. We have taken the comments and suggestions into consideration and submitted a revised version of the paper. Most changes were made in Section 5.2 and Appendix B (a revised version of the supplementary material contains a diff that highlights all changes).

Before addressing individual comments, we wanted to address a shared concern about having results in environments with high dimensional state/observation spaces. We completely agree that doing so is an important next step for this work but respectfully submit that the paper is complete as-is. Our main focus in this paper was to explore the distinction between on-policy data and on-policy sampling within the context of policy evaluation and show how on-policy data can be obtained more efficiently with off-policy sampling. We believe this work opens up an exciting new direction for on-policy RL algorithms and as an initial paper in this direction, we chose to include a mixture of theoretical and empirical results to provide a solid foundation for future work. Also, please note that our empirical analysis (including additional results in appendices) is still fairly extensive with a variety of ablations and sensitivity analyses.

---

### Meta-Review · Area_Chair_LNvN · 2022-08-26

**Recommendation:** Accept
**Confidence:** Less certain

**Metareview:**

The paper seeks to improve the efficiency of policy evaluation in reinforcement learning settings. Monte-Carlo sampling is commonly used for on-policy evaluation, but the paper eloquently shows that non-iid off-policy sampling can be a better strategy. The reviewers agreed that the technique was novel, interesting and promising to share with the community.

Some reviewers questioned the theoretical analysis that the authors clarified during the feedback phase. To strengthen the paper, the authors should experiment with ROS in richer problem domains (e.g., with function approximators) to identify realistic regimes where ROS helps empirically. For instance, if ROS is made easy enough to use with DeepRL policies in complex domains, it could be a very significant contribution to on-policy evaluations that are currently plagued by high variance from Monte-Carlo sampling.

**Award:**

No

---

### Decision · Program_Chairs · 2022-09-14

Accept